# Concerning the stability of seawater electrolysis: a corrosion mechanism study of halide on Ni-based anode

Sixie Zhang [1,2,3,5], Yunan Wang[1,2,3,5], Shuyu Li[3,4], Zhongfeng Wang[1,2,3], Haocheng Chen[1,2], Li Yi[1,2], Xu Chen [1,2], Qihao Yang [1,2], Wenwen Xu [1,2] ✉, Aiying Wang [3,4] & Zhiyi Lu [1,2,3] ✉

The corrosive anions (e.g., $Cl^-$) have been recognized as the origins to cause severe corrosion of anode during seawater electrolysis, while in experiments it is found that natural seawater (~0.41 M $Cl^-$) is usually more corrosive than simulated seawater (~0.5 M $Cl^-$). Here we elucidate that besides $Cl^-$, $Br^-$ in seawater is even more harmful to Ni-based anodes because of the inferior corrosion resistance and faster corrosion kinetics in bromide than in chloride. Experimental and simulated results reveal that $Cl^-$ corrodes locally to form narrow-deep pits while $Br^-$ etches extensively to generate shallow-wide pits, which can be attributed to the fast diffusion kinetics of $Cl^-$ and the lower reaction energy of $Br^-$ in the passivation layer. Additionally, for the Ni-based electrodes with catalysts (e.g., NiFe-LDH) loading on the surface, $Br^-$ causes extensive spalling of the catalyst layer, resulting in rapid performance degradation. This work clearly points out that, in addition to anti-$Cl^-$ corrosion, designing anti-$Br^-$ corrosion anodes is even more crucial for future application of seawater electrolysis.

Green hydrogen ($H_2$) production from the electrolysis of seawater in an alkaline environment, which eliminates the complex purification process and suppresses the undesirable chlorine evolution reaction (ClER), has been widely acknowledged as an economical and sustainable technology that can timely make full use of the excess renewable energy from offshore[1–6]. The commonly agreed intractable issue in this system is the poor durability of anodes, which is mainly attributed to the severe $Cl^-$-induced corrosion[7–9]. Indeed, contemporary works have demonstrated that the stability of anodes can be improved by designing active catalysts with an anti-$Cl^-$ corrosion layer, such as transition-metal sulfides[10–12], nitrides[13–15], phosphides[16–19], etc. However, after scrutinizing the previous studies[13,16,20–23], an interesting phenomenon comes out that the stability of Ni-based anodes tested in

alkaline seawater is obviously shorter than that in alkaline saline water (0.5 M NaCl, Supplementary Fig. 1), also consistent with our tested results (Supplementary Fig. 2). These unexpected results herald that, besides $Cl^-$, other complex chemical compositions in seawater will also shorten the lifetime of the anode.

Considering that a positive voltage was applied on the anode during electrolysis, the suspect aggressive ions are focused on anions in seawater. Based on publicly available data[24] and our tested results (Supplementary Table 1), the content of anions in seawater follows the sequence of $Cl^- > SO_4^{2-} > HCO_3^- > Br^-$. Previous works have demonstrated that the addition of oxyanions (e.g., $SO_4^{2-}$ and $HCO_3^-$) to alkaline electrolytes could hardly affect the stability of the anode[25] and even alleviate the attack from $Cl^-$[21]. Accordingly, $Br^-$, ranked fourth

[1]Key Laboratory of Advanced Fuel Cells and Electrolyzers Technology of Zhejiang Province, Ningbo Institute of Materials Technology and Engineering, Chinese Academy of Sciences, Ningbo 315201 Zhejiang, P. R. China. [2]Qianwan institute of CNITECH, Ningbo 315201 Zhejiang, P. R. China. [3]University of Chinese Academy of Sciences, Beijing 100049, P. R. China. [4]Key Laboratory of Marine Materials and Related Technologies, Ningbo Institute of Materials Technology and Engineering, Chinese Academy of Sciences, Ningbo 315201 Zhejiang, P. R. China. [5]These authors contributed equally: Sixie Zhang, Yunan Wang. ✉e-mail: xuwenwen@nimte.ac.cn; luzhiyi@nimte.ac.cn

place, possibly is the predominant anion that aggravates the corrosion of anodes. It should be noted that although the original concentration of Br⁻ in seawater is quite low (around 0.53 mM), it will be gradually accumulated to ~0.5 M with the continuous replenishment of seawater during one year of electrolysis (see details in the Supplementary Information, Supplementary Fig. 3). However, the critical role of Br⁻ has not been noticed before and remains elusive. Thus, it is necessary and urgent to refine the corrosion mechanism of halides (Cl⁻ and Br⁻) on the conventional anode and identify the characteristics of respective corrosion processes.

Herein, the corrosion behaviors of three Ni substrates (Ni foil, Ni foam, and Ni mesh) in Cl⁻- and Br⁻-based electrolytes are systematically investigated experimentally and theoretically, where we notice that Br⁻ is much more aggressive than Cl⁻ in terms of the critical corrosion descriptors. Assessment of cyclic polarization curves (CPC) demonstrates that Ni substrates exhibit lower pitting potential ($E_{pit}$) and faster corrosion kinetics in Br⁻-containing electrolyte (BrE) compared with those in Cl⁻-containing electrolyte (ClE). Further scanning vibrating electrode technique (SVET) and in situ Raman experiments elucidate that Cl⁻ tends to corrode Ni substrates locally to form narrow-deep pits while Br⁻ prefers to etch extensively to generate wide-shallow pits, which can be explained by the easier reaction of Br⁻ and the fast diffusion of Cl⁻ according to the theoretical simulations. In addition, the different Cl⁻ and Br⁻ corrosion mechanism triggers huge discrepancy in the corrosion behaviors of Ni-based electrodes with catalysts loading on the surface, where Cl⁻ causes a few pits while Br⁻ leads to the extensive exfoliation of catalyst layer after a period of electrolysis. Our corrosion mechanism exploration of Cl⁻ and Br⁻ will be of great guiding significance in synthesizing of anodes for durable seawater electrolysis.

## Results and discussion

### Aggressiveness of halides on Ni substrates

During seawater splitting, pitting corrosion, which commonly occurs on metal-based anodes, is a localized metal dissolution process triggered by aggressive halides[26–29]. Typically, CPC (Supplementary Fig. 4), which directly displays the relationship between the current (log |$j$|) and voltage ($E$), is generally employed in evaluating the anti-pitting ability of anode materials under the electric field environment[30–32] (see details in Supplementary Information). Three commercial Ni-based materials (Ni foil, Ni foam, and Ni mesh) with spontaneously formed dense NiO passive film (Supplementary Figs. 5–8) were selected as anodes to evaluate the corresponding anti-pitting performance by CPC method in alkaline solutions containing 0.5 M NaCl and NaBr, respectively. Notably, the concentration of NaOH was kept low (1 mM, pH-10.8) to avoid signal interference from both oxygen evolution reaction (OER) and Ni²⁺/Ni³⁺ redox reaction (Supplementary Fig. 9). The assessment results (Fig. 1a–c, Supplementary Fig. 10, and Supplementary Table 2) demonstrated that all Ni anodes suffered from severe corrosion in BrE, with an easier pitting trend (i.e., lower $E_b$ and $E_{pit}$)[33–35] and faster pitting kinetics (i.e., smaller Tafel slope_pit) than those in ClE. Taking Ni foil as a representative anode, the $E_b$ and $E_{pit}$ values measured in BrE were negatively shifted ~88 mV and ~32 mV compared with those in ClE. Simultaneously, the Tafel slopes_pit of Ni foil in BrE and ClE were calculated as 80.1 mV dec⁻¹ and 137.4 mV dec⁻¹, respectively. Similar trends were observed for Ni foam and Ni mesh in the corresponding CPC evaluations, further validating that Br⁻ was more aggressive in pitting corrosion of Ni substrates.

Further, the operando electrochemical impedance spectroscopy (EIS) under working conditions[36,37] was utilized to evaluate the dynamic

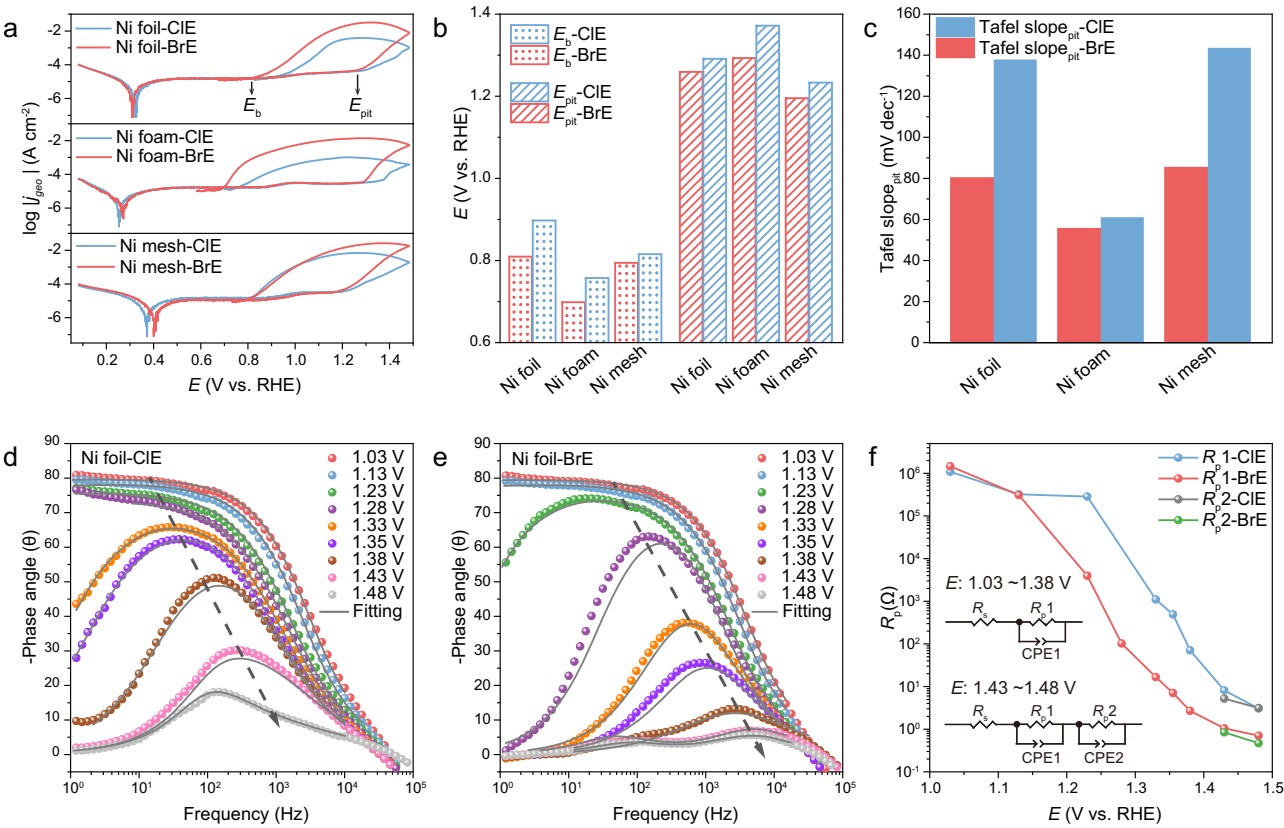

**Fig. 1 | Anti-pitting ability assessments of Ni substrates in ClE and BrE. a** CPCs of Ni substrates tested in ClE and BrE at a scan rate of 10 mV s⁻¹. **b, c** Comparison of $E_b$, $E_{pit}$, and Tafel slope_pit values obtained from CPCs. **d, e** Bode-phase plots of Ni foil at different potentials in ClE and BrE, respectively. **f** Corresponding equivalent resistances ($R_{p1}$ and $R_{p2}$) and potentials of Ni foil at different potentials in two electrolytes, where $R_{p1}$ and $R_{p2}$ are the polarization resistances at different frequency regions. The individual equivalent circuit models for different potentials are embedded in the diagram.

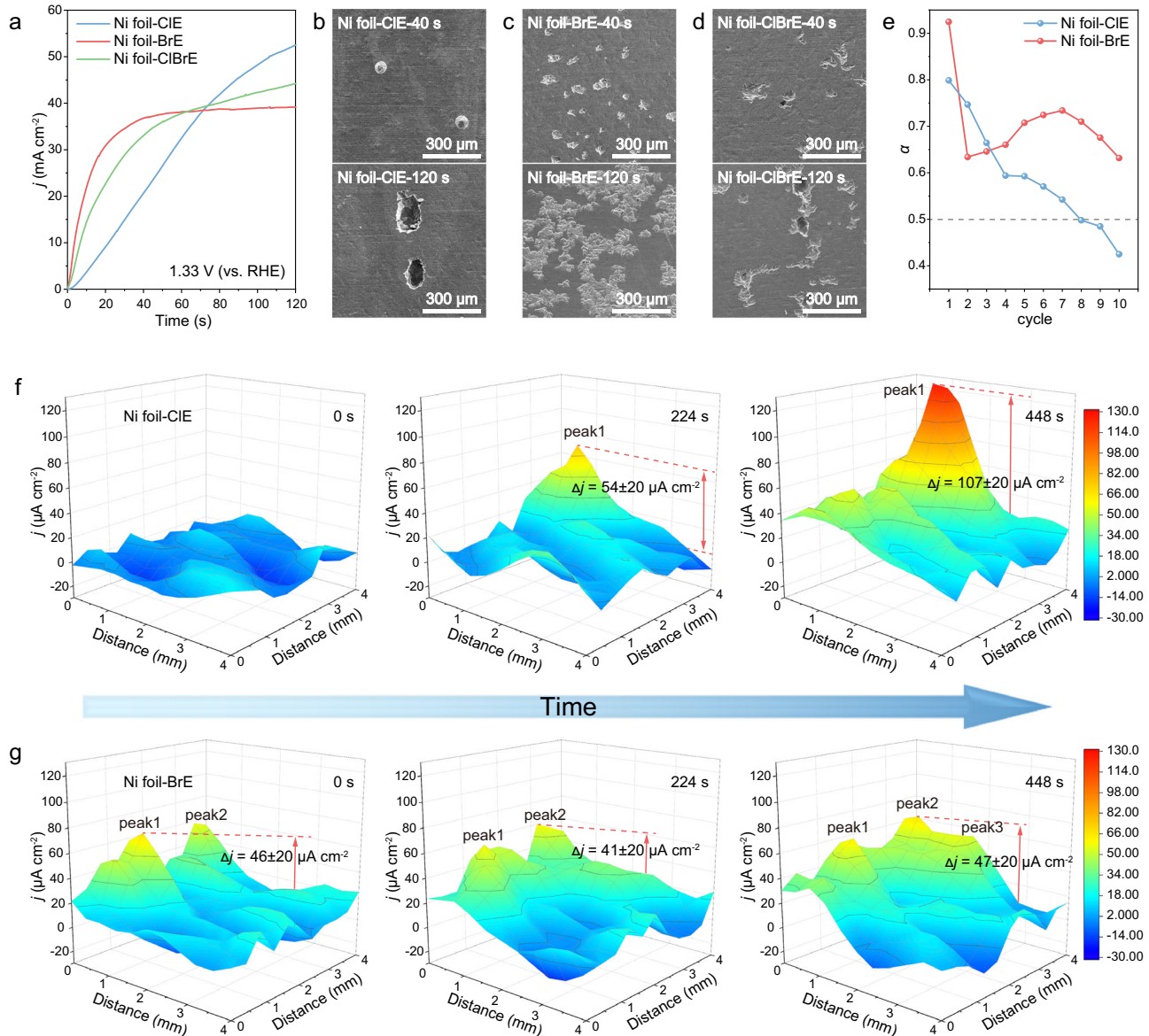

**Fig. 2 | Corrosion behaviors of Ni foil in halide-containing electrolytes.**
**a** Potentiostatic polarization curves of Ni foil tested in ClE and BrE at a potential of 1.33 V vs. RHE. **b**–**d** Corresponding morphology of pits on Ni foil corroded by $Cl^-$ and $Br^-$, and both $Cl^-$ and $Br^-$ during the potentiostatic polarization tests. **e** $\alpha$-cycle plots obtained by fitting the slope of linear part in the high frequency region in time-dependent imaginary part of the impedance-frequency plots. **f, g** Changing trends of SVET current density maps with time in ClE and BrE, respectively.

corrosion resistance of Ni substrate in both aforementioned BrE and ClE (Supplementary Fig. 11), respectively. As shown in Fig. 1d and e, the phase angle at the middle-frequency region began to decrease and shift at a threshold potential, indicating an oxidation process of Ni metal (i.e., corrosion of Ni, $Ni^0 \rightarrow Ni^{2+}$, Supplementary Fig. 12)[38,39]. Notably, the threshold potential in BrE (1.23 V) was inferior to that in ClE (1.33 V), indicating that Ni foil was easier to be corroded by $Br^-$ than $Cl^-$, consistent with the CPC analysis. To further quantify the physical processes that occurred in the system, the EIS data were fitted with equivalent circuit models (Fig. 1f and Supplementary Fig. 13)[40,41], and the corresponding optimum fit parameters were shown in Supplementary Table 3. For Ni foil in BrE, below 1.23 V, the equivalent polarization resistance $R_p1$ was large, indicating the charge transfer was exceedingly weak. After reaching 1.23 V, the $R_p1$ significantly decreased, reflecting the $Br^-$ corroding-induced electrooxidation of $Ni^0$ to $Ni^{2+}$. When the potential exceeded 1.43 V, the equivalent resistance $R_p2$ appeared, which subsequently decreased with the increase of potential,

suggesting that $Ni^{2+}$ was oxidized to $Ni^{3+}$. A similar tendency was observed in ClE, while the decrease of $R_p1$ was posterior to that in BrE, proving the weaker aggressiveness of $Cl^-$ to Ni foil. It is worth noting that the values of $R_p1$ and $R_p2$ for Ni foil in BrE were smaller than that in ClE, further verifying the faster pitting kinetics in BrE.

## Pitting behaviors of Ni substrates in halide-containing electrolytes

To further investigate the individual pitting behavior of Ni foil in ClE and BrE, we recorded the *I-t* curves and surface morphology evolution of the anodes under the same operating condition. The working potential was 1.33 V vs. RHE, at which the current can be attributed to the corrosion of Ni substrate. As shown in Fig. 2a, the current of Ni foil in ClE was gradually increasing, while the BrE triggered an initial fast current increase followed by a stable corrosion current density (~40 mA $cm^{-2}$). Note that when both $Cl^-$ and $Br^-$ were present in the electrolyte (ClBrE), the current of Ni foil increased fast initially and then

gradually. Moreover, the corresponding SEM and in situ optical microscopy images demonstrated completely different corrosion behaviors of Ni substrate in ClE, BrE and ClBrE. In ClE, the surface of Ni exhibited merely several tiny penetrating pits, and afterwards the existing penetrating pits became larger and deeper with the extension of reaction time (Fig. 2b, Supplementary Fig. 14a, and Supplementary Movie 1). In sharp contrast, in BrE, multiple craters with wide and shallow shapes were formed on Ni foil surface, and both the number and area of the pits increased with the time extension (Fig. 2c, Supplementary Fig. 14b, and Supplementary Movie 2). Notably, in ClBrE, the Ni foil exhibited a mixed corrosion behavior with both deep and shallow penetrating pits (Fig. 2d). This phenomenon was possibly attributed to the initial corrosion of Br⁻ and continue attack from Cl⁻. Moreover, the constant phase element exponent ($\alpha$) values calculated from operando EIS spectra (Fig. 2e, Supplementary Fig. 15, and Supplementary Table 4) in ClE and BrE exhibited different trends, also demonstrating the different corrosion behaviors of Ni foil in BrE and ClE (see details in the Supplementary Information)[42–44].

SVET, a powerful instrument for evaluating the current density distributions and variations during corrosion processes[45–48], was also utilized to investigate the corrosion behavior of Ni foil at a local scale. For the Ni foil in ClE, a negligible reaction signal was observed on the SVET map at the initial stage, as shown in Fig. 2f. With the reaction proceeding, a sharp current peak appeared and continuously grew up from -54 μA cm⁻² to -107 μA cm⁻², implying the occurrence and facilitation of pit corrosion in the same location. In contrast, the current density map of Ni foil in BrE exhibited a multi-site simultaneous reaction mode, as shown in Fig. 2g. As time went on, the peak current was only slightly varied between 41 and 47 μA cm⁻², while the peak area expanded significantly, in agreement with our SEM results.

## Pitting mechanism of Ni substrates in halide-containing electrolytes

During the past decades, a number of theoretical models have been proposed to describe initiation processes leading to passive film breakdown and pitting corrosion, mainly including (1) adsorption-induced mechanism, (2) ion migration and penetration model, and (3) mechanical film breakdown theory (Supplementary Fig. 16, see details in Supplementary Information)[49–57]. Generally, different mechanisms or combinations of these mechanisms are suitable for different corrosion systems[50,57]. Note that the mechanical film breakdown usually caused by mechanical factors, which was less relevant to the halide species. Thus, in our case, we propose that the corrosion mechanism of Ni substrates is a combination of (1) and (2), and the quasi in situ Raman spectra results may satisfy the conclusion (Supplementary Fig. 17). The discrepancies in corrosion behavior can be attributed to the different contributions of adsorption and penetration processes. Therefore, we conducted the density functional theory (DFT) simulations to investigate the adsorption process and nudged elastic band (NEB) calculations to study the penetration process (Fig. 3a).

Based on the adsorption-induced mechanism[51,52], we thus proposed the possible reactions during the corrosion process:

$$\text{Passivation process}: Ni + 2OH^- \rightarrow NiO + 2e^- + H_2O \tag{1}$$

$$\text{Replacement process}: NiO + 2yX^- + yH_2O \rightarrow NiO_{1-y}X_{2y} + 2yOH^- \ (0{<}y{<}1) \tag{2}$$

$$NiO_{1-y}X_{2y} + (2-2y)X^- + (1-y)H_2O \rightarrow NiX_2 + (2-2y)OH^- \tag{3}$$

$$\text{Precipitation process}: 2OH^- + NiX_2 \rightarrow NiOH_2 + 2X^- \tag{4}$$

Where X⁻ denotes Cl⁻ and Br⁻.

Subsequently, the corrosion thermodynamics of NiO in halide were studied by DFT simulations, and the simulated results (Fig. 3b and Supplementary Fig. 18) showed that the detachment of Ni-X from intermediates NiO(ov)-X (NiO with oxygen displaced by halide) was the rate-determining step (RDS) in both ClE and BrE. Notably, the free energy of RDS in BrE (2.55 eV) was inferior to that in ClE (2.97 eV), suggesting that NiO was prone to be corroded by Br⁻, which was consistent with the CPC results.

As for the penetration process, we employed the NEB calculations to investigate the diffusion kinetics of halides from the surface to the NiO lattice. According to the NEB results (Fig. 3c, d), the Cl⁻ and Br⁻ experienced the same penetration process from the NiO surface to the lattice, and the energy barrier to overcome to reach the final state was lower for Cl⁻ (3.40 eV) than that for Br⁻ (3.91 eV), implying that Cl⁻ would diffuse easier into the lattice interior.

Combined with the experimental and theoretical results, the specific pitting process of Ni substrates in ClE, BrE, and ClBrE can be illustrated as follows. For the Ni anode in ClE, initially, the surface reaction is dominated by the oxidation of Ni, which mainly causes the thickening of the passive layer. With the increase of the dissolution rate, the pits will nucleate on the passive film. Since Cl⁻ is difficult to react with the surface NiO film but easily diffuses into the NiO lattice, thus the Cl⁻-induced corrosion is more likely to react in the vertical direction (Fig. 4a), resulting in the formation of narrow and deep pits on Ni surface. As for the Ni anode in BrE, the Br⁻ that gathers on the anode surface will preferentially undergo corrosion reactions at multiple sites rather than penetrate into the lattice, and thus the Br⁻-induced pitting tends to react horizontally, resulting in wide-shallow pits (Fig. 4b). For the Ni anode in ClBrE, Br⁻ may react with the surface NiO first, followed by the diffusion and corrosion of Cl⁻, forming a mixed corrosion morphology (Fig. 4c).

## Corrosion behaviors of catalysts/Ni anodes in halide-containing electrolytes

As the most popular and excellent catalyst for OER, NiFe-LDH directly grown on Ni foam (NiFe-E, Supplementary Fig. 19) was selected as the typical example to explore the corresponding Cl⁻-induced and Br⁻-induced corrosion behaviors. The CPC (Fig. 5a and Supplementary Fig. 20) results demonstrated that the NiFe-E possessed lower $E_{pit}$ and Tafel slope$_{pit}$ in BrE than ClE, indicating that NiFe-E has a more severe corrosion in BrE, consistent with the CPC results for Ni substrates.

Further, the durability of NiFe-E was tested under the practical conditions in alkaline saline water (1 M NaOH + 0.5 M NaCl, 1 M NaOH + 0.5 M NaBr, and 1 M NaOH + seawater electrolytes) at a current density of 400 mA cm⁻² (Fig. 5b and Supplementary Fig. 21), and the morphologies of NiFe-E after the voltage fluctuations were recorded. For NiFe-E in Cl-based electrolyte, the SEM morphology (Fig. 5c) revealed that the Cl⁻-induced pit preferentially occurred in the weak sites of the catalyst layer and easily caused deep corrosion in the same place. The insert schematic diagram of Cl⁻-induced corrosion behavior of NiFe-E emphasized the importance of high coverage of catalyst/anti-halide layer. In contrast, only exfoliation of the catalyst layer can be observed in Br-based electrolyte (Fig. 5d), indicating that Br⁻ was more aggressive than Cl⁻ even in high alkali concentration. The schematic diagram of Br⁻-induced corrosion behavior of NiFe-E illustrated the criticality of enhancing the connection between catalyst layer and the substrate. As for NiFe-E in seawater, both the deep pits and spalling of catalyst layer can be observed (Fig. 5e), indicating the mixed corrosion behavior. Thus, even though Br⁻ is of trace amount (0.53 mM) in seawater, its influence on the corrosion behavior of Ni substrates should be attracted more attention. Furthermore, we also observed the similar corrosion behaviors of anodes with different catalyst layers (pure Ni foil, NiCo-LDH, NiFeP, and NiFe-H) in Cl⁻ and Br⁻ containing electrolytes (Supplementary Figs. 22–26), indicating that Br⁻ corrosion

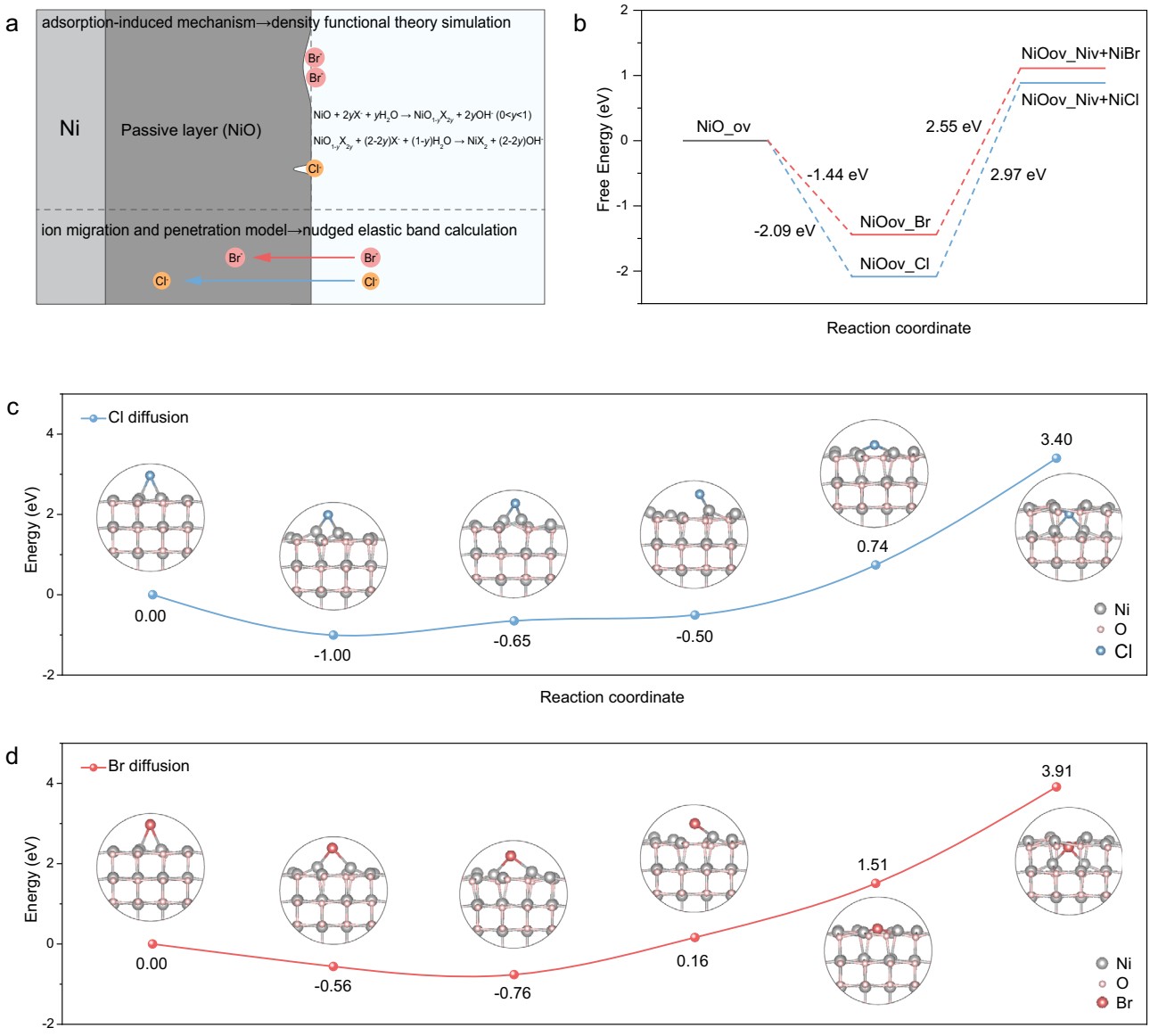

**Fig. 3 | Corrosion mechanism exploration of Ni-substrate in halide-containing electrolytes. a** Schematic diagram of the subsequent theoretical calculations corresponding to the two pitting initiation mechanisms. **b** Calculated free-energy diagrams for NiO corroded by halide, where ov denoted the oxygen vacancy and Niv denoted the Ni vacancy. **c**, **d** Activation energies simulated by NEB calculations for the diffusion processes of Cl⁻ and Br⁻ in a NiO host.

induced spalling of the catalyst layer was a common and general phenomenon in integrated electrodes (substrate + catalyst on surface).

In summary, aiming at understanding the corrosion mechanism of anode in seawater splitting, we revealed the different Cl⁻-induced and Br⁻-induced pit corrosion processes of Ni-based anodes. It is elucidated that Br⁻ was more aggressive to the anodes. The CPC evaluations demonstrated that Ni substrates (Ni foil, Ni foam, and Ni mesh) possessed lower $E_b$ and $E_{pit}$ and faster corrosion kinetics in BrE than in ClE. With the assistance of in situ optical microscopy and SVET, we recorded the Cl⁻-induced, Br⁻-induced, and Cl⁻-Br⁻-co-induced corrosion behaviors, where Cl⁻-induced corrosion led to narrow-deep pits, Br⁻-induced corrosion caused the appearance of wide-shallow pits, and Cl⁻-Br⁻-co-induced corrosion resulted in the mixed pit morphologies. Further DFT and NEB simulation results explained the special pit corrosion behaviors from the thermodynamic and kinetics aspects, where Cl⁻ was preferred to diffuse into the NiO lattice and Br⁻ reacted more easily with NiO. In addition, the performance of Ni-based electrodes with catalysts (e.g., NiFe-LDH)

loading on the surface illustrated that the Br⁻-induced pit corrosion could peel the catalyst layer off the substrate, causing the quick breakdown of the anode. According to the results, we believed that the design of robust anodes with anti-Br⁻ corrosion performance is vital to achieve highly stable seawater electrolysis in the future.

## Methods

### Preparation of Ni foil electrode

Ni foil (99.99%, thickness is 0.15 mm, purchased from Tan Qian Lang) was trimmed into 1 cm × 2 cm pieces. And then, these pieces were cleaned to degreasing via sonication in acetone, ethanol, and deionized water for 10 min, respectively. Finally, the Ni foil was dried at 60 °C for 3 h.

### Preparation of Ni foam electrode

The preparation process of Ni foam (99.99%, thickness is 1.6 mm, the porosity is 90 ppi, purchased from Suzhou sinero technology Co., LTD.) electrode was the same as Ni foil.

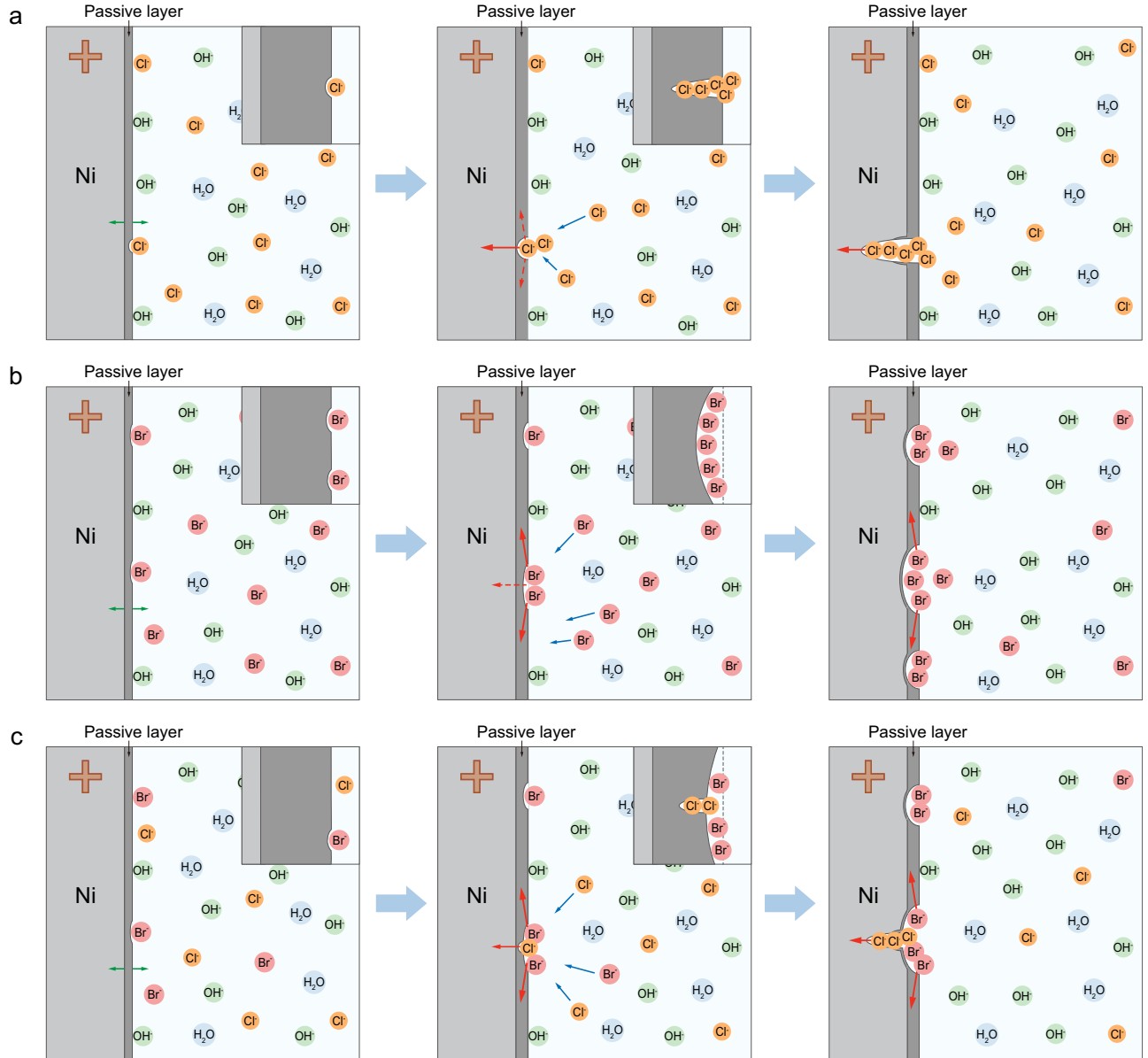

**Fig. 4 | Corrosion processes of Ni foil in ClE and BrE. a–c** Schematic illustrations of Ni foil corroded by Cl⁻, Br⁻, and both Cl⁻ and Br⁻, respectively. Insets are detailed views of NiO passive film corroded by halide.

### Preparation of Ni mesh electrode

The preparation process of Ni mesh (99.99%, thickness is 0.5 mm, 46 mesh, purchased from Suzhou sinero technology Co., LTD.) electrode was the same as Ni foil.

### Synthesis of NiFe-H electrode

Ni foam (99.99%, thickness is 1.6 mm, the porosity is 90 ppi, purchased from Suzhou sinero technology Co., LTD.) was sonication degreased in acetone, ethanol, and deionized water for 10 min, then cleaned with 2 M $H_2SO_4$ in an ultrasound bath for 15 min to remove the surface NiO layer, and finally washed with deionized water until the pH was neutral. 0.5 mmol of $Ni(NO_3)_2$, 0.5 mmol of $Fe(NO_3)_3$, and 10 mmol of urea, were dissolved in 35 mL of deionized water. Then, the aqueous solution and the acid-washed Ni foam were transferred to Teflon-lined stainless autoclave (50 mL) and maintained at 120 °C for 12 h. Finally, the electrode was washed with water several times and dried at 60 °C for 2 h.

### Synthesis of NiFe-E electrode

The NiFe-E electrode was synthesized by an electrochemical deposition method in a typical three-electrode system. Ni foam (99.99%, thickness is 1.6 mm, the porosity is 90 ppi, purchased from Suzhou sinero technology Co., LTD.) was sonication degreased in acetone, ethanol, and deionized water for 10 min, then cleaned with 2 M $H_2SO_4$ in an ultrasound bath for 15 min to remove the surface NiO layer, and finally washed with deionized water until the pH was neutral. The above-mentioned acid-washed Ni foam served as the working electrode, Pt foil acted as the counter electrode, and Ag/AgCl (saturated KCl) worked as the reference electrode. The electrodeposition solution contained 6 mM $Ni(NO_3)_2$ and 2 mM $Fe(NO_3)_3$, and was kept at 10 °C by cold water bath. The working electrode was held at −1 V vs. Ag/AgCl (saturated KCl) for 20 min. After the electrodeposition process, the electrode was washed with water several times and dried at 60 °C for 2 h.

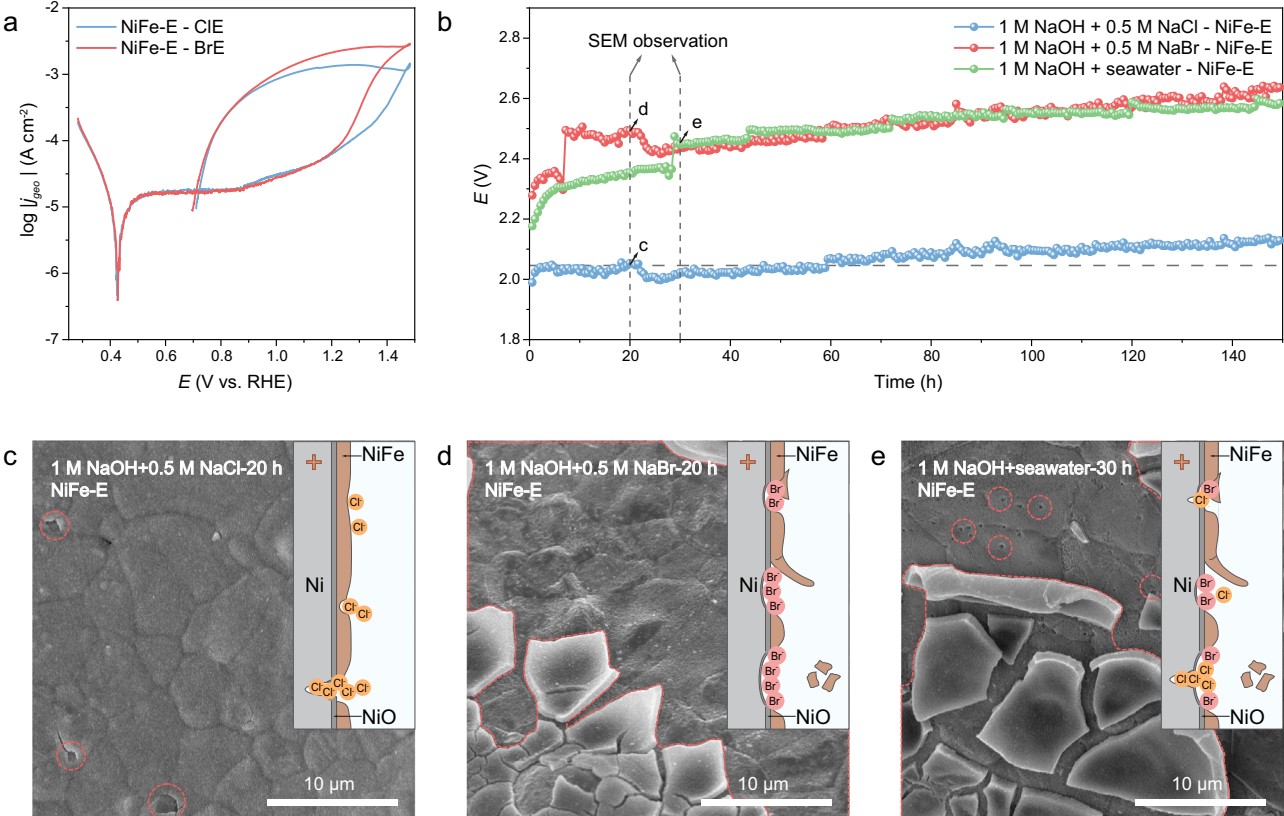

**Fig. 5 | Corrosion of NiFe-E in ClE, BrE, and seawater. a** CPCs of NiFe-E in ClE and BrE. The scan rate was 10 mV s⁻¹. **b** Durability tests of NiFe-E at a current density of 400 mA cm⁻² in 1 M NaOH + 0.5 M NaCl, 1 M NaOH + 0.5 M NaBr, and 1 M NaOH + seawater electrolytes. **c–e** Corresponding SEM images of NiFe-E observed after 20 h operation in 1 M NaOH + 0.5 M NaCl, 20 h operation in 1 M NaOH + 0.5 M NaBr, and 30 h operation in 1 M NaOH + seawater.

## Synthesis of NiCo-LDH electrode

0.6 mmol of $Ni(NO_3)_2$, 0.2 mmol of $Fe(NO_3)_3$, 3 mmol $CH_4N_2O$ and 3 mmol $NH_4F$, were dissolved in 30 mL of deionized water. Then, the aqueous solution and the acid-washed Ni foam were transferred to Teflon-lined stainless autoclave (50 mL) and maintained at 140 °C for 4 h. Finally, the electrode was washed with water several times and dried at 60 °C for 2 h.

## Synthesis of NiFeP electrode

Hydrothermal-NiFe-LDH nanoarrays precursor and 0.1 g $NaH_2PO_2 \cdot H_2O$ were placed at two sides of the porcelain boat in a tube furnace and then heated at 300 °C for 10 min in $N_2$ atmosphere.

## Material characterizations

The morphologies of the samples were observed by field emission Hitachi S-4800 scanning electron microscope (SEM). X-ray photoelectron spectroscopy (XPS) measurements were carried out by using an ESCALAB 250 model. Raman measurements were conducted by using a RENISHAW inVia Raman spectrometer. The content of anions in seawater was measured by Thremo ICS1100 ion chromatography (IC).

## Electrochemical measurements

The cyclic polarization curves (CPC), electrochemical impedance spectroscopy (EIS), and potentiostatic polarization curves were measured on electrochemical workstation (ChenHua CH instrument 760E) at room temperature with a three-electrode system, where the test samples served as the working electrode, Pt foil (1 cm × 1 cm) acted as the counter electrode, and the saturated calomel electrode (SCE) was the reference electrode. And the 1 mM NaOH + 0.5 M NaCl (ClE), 1 mM NaOH + 0.5 M NaBr (BrE), and 1 mM NaOH + 0.25 M NaCl + 0.25 M NaBr (ClBrE) solutions (pH ~10.8) were selected as the electrolytes. The test samples were clamped by a 100%-Ti holder to make electrical contact, and their areas immersed in the electrolyte were controlled at 1 cm². The CPCs were obtained at a scan rate of 1 mV s⁻¹. The EIS tests were conducted at a perturbing AC amplitude of 5 mV and scanning frequencies of 10 kHz to 1 Hz. As for potentiostatic polarization tests, 1.38 V vs. RHE voltage was applied on the test samples for 120 s.

## Scanning vibrating electrode technique (SVET) measurements

The local current density mappings at Ni foil surface during the potentiostatic polarization tests were measured by the VersaScan SVET module (Supplementary Fig. 27) from Princenton Applied Research (USA) using a Pt/Ir probe with a diameter of 10 μm. The tip-substrate distance was <100 μm. The probe vibrated along the Z direction with a vibration frequency of 80 Hz and an amplitude of 30 μm. The scan area was 4 mm × 4 mm and 11 × 11 points along X and Y axes. The obtained potential signals were converted to the local current density (J) by using Ohm's law, as shown in the following equation[48,58]:

$$J = -\Delta\phi\frac{k}{A}(A/cm^2) \tag{5}$$

Where $\Delta\phi$ is the electric potential drop, $k$ is the electrolyte conductivity (-2.5 S m⁻¹), and $A$ is the vibration amplitude (30 μm). For the time-dependent SVET measurement, the applied voltage was set as 1.3 V vs. RHE, and the probe scanning cycles were continuous. Completing one map took 222 s, including a 5 s delay time and a 1 s restitution time. The time interval between each circle was 2 s, during which the data file was saved and named.

## In situ optical microscopy characterizations

The corrosion behaviors of Ni foil in ClE and BrE during the potentiostatic polarization tests were recorded by the optical contact angle and surface/interfacial tension measuring system (OSA 60 G, LAUDA Scientific). Xenon lamp was selected as the light source, and quartz-built Photoelectrochemical cell was used to ensure the light transmission.

## Electrochemical in situ Raman measurements

Electrochemical in situ Raman measurements were conducted by using a RENISHAW inVia Raman spectrometer equipped with a Leica TCS SP8 CARS microscope and a Spectra-Physics 532 nm Ar laser. Acquisition time for each spectrum was 3 s with sweeps from 300 to 1000 cm$^{-1}$. All the measurements were performed in a custom-made Spectro-electrochemical cell at room temperature. The three-electrode configuration was used, where Ni foil was performed as the working electrode, a Pt wire was set as the counter electrode, and the SCE was the reference electrode. The electrolytes included 1 mM NaOH + 0.5 M NaCl and 1 mM NaOH + 0.5 M NaBr. The potentiostatic polarization method was employed with voltage of 1.33 V vs. RHE.

## Theoretical computation details

All the DFT calculations are performed by the Vienna Ab initio Simulation Package (VASP)[59] with the projector augmented wave (PAW) method[60]. The exchange-functional is treated using the generalized gradient approximation (GGA) with Perdew-Burke-Ernzerhof (PBE)[61] functional. The energy cutoff for the plane wave basis expansion was set to 400 eV. Partial occupancies of the Kohn−Sham orbitals were allowed using the Gaussian smearing method and a width of 0.2 eV. The structure of Ni, NiO, NiO(ov) were built. A climbing image nudged elastic band (CI-NEB)[62] method was used to locate the transition states. The Brillouin zone was sampled with Monkhorst mesh of $3 \times 3 \times 1$ for the optimization for all the structures. The self-consistent calculations apply a convergence energy threshold of $10^{-5}$ eV, and the force convergence was set to 0.05 eV/Å for the structure optimization.

# Data availability

The data that support the findings of this study are available from the corresponding author upon reasonable request.

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

## Acknowledgements

This work was supported by the National Key Research and Development Project (2021YFA1502200/Z.Y.L.), Ningbo Yongjiang Talent Introduction Programme (2021A-036-B/Z.Y.L.), Bellwethers Project of Zhejiang Research and Development Plan (2022C01158/Z.Y.L.), the Ningbo S&T Innovation 2025 Major Special Program (2022Z205/Q.H.Y. and 2020Z107/W.W.X.), National Natural Science Foundation of China (NSFC; NO. 22105214/W.W.X. and NO. 52201285/X.C.), and Natural Science Foundation of Ningbo (20221JCGY010295/W.W.X.).

## Author contributions

Z.L., W.X., Y.W., and S.Z. designed the project and wrote the manuscript; S.Z., S.L., Z.W., H.C., L.Y. carried out the experiments and collected the data; Z.L., W.X., Y.W., A.W., X.C., Q.Y., and S.Z. analyzed data. All authors discussed the results and commented on the manuscript.

## Competing interests

The authors declare no competing interests.
