## [Peer Review File · Nature Communications]

Concerning the stability of seawater electrolysis: a corrosion mechanism study of halide on Ni-based anodeREVIEWER COMMENTS

Reviewer #1 (Remarks to the Author):

Seawater electrolysis in alkaline environment is a reasonable and scalable pathway in seawater splitting field due to its low energy consumption, high selectivity and compatibility with current alkaline electrolyser. However, as the electrolyte should be reused, the accumulation of anions in electrolyte will definitely lead to severe electrode corrosion. It is well-known that Cl⁻ in seawater attacks the electrode, while the authors in this manuscript highlight that the accumulated Br⁻ in seawater is even more serious in terms of the corrosion behavior. Based on the detailed corrosion mechanism on conventional Ni substrate, it is quite convincing that Br⁻ corrosion should be taken into consideration in the future study of alkaline seawater splitting. This is an interesting study, which is of both fundamental and practical significance for the application of seawater splitting. Thus, I recommended this work to be accepted after minor revision:

1. The authors conducted the experiments in 0.5 M NaCl and 0.5 M NaBr. Although the difference is obvious, it cannot directly reflect the corrosion of natural seawater, in which the concentration of Br⁻ is significantly lower than that of Cl⁻. Thus, I suggest the authors test the corrosion behavior of natural seawater on Ni substrate to clarify the role of trace amount of Br⁻.
2. In this work, the corrosion behavior of NiFe-LDH nanoarrays electrode in Br⁻ and Cl⁻ has been carefully investigated. What about other catalytic materials?
3. I also want to know the different corrosion behaviors of the same catalyst in Cl⁻ and Br⁻ with varied morphologies.
4. At the beginning of this paper, the corrosion behavior of catalysts with the same concentration (0.5 M) of Cl⁻ and Br⁻ in 1 mM NaOH electrolyte was investigated. However, in Figure 5, the electrolyte used is 1 M NaOH. It's a little confusing. Is the conclusion that Br⁻ leads to more severe corrosion also applicable to seawater with higher alkali concentrations?
5. How did the authors determine computational model? Any evidence from experiments?
6. It is suggested that the corrosion processes of Ni under the electrolyte with both Cl⁻ and Br⁻ should be illustrated. Please demonstrate it in Fig. 4.

Reviewer #2 (Remarks to the Author):

The work under consideration maybe regarded of significance to the fields of corrosion and energy generation using electrochemical methods. In particular, the enhanced corrosion rates experienced by electrodes in marine environment compared to simulated laboratory tests is certainly a hot topic. But the following issues need addressing:

(1) The Authors in this work claim the neglected contribution of bromide ions to be the main source for such discrepancy, and they justify such effect due to the increased amount of salts at the electrode compartment due to the operation of the electrochemical cell as the rationale for investigating the effect of bromide ions.

Accordingly, a study is presented by replacing the typical concentration of chloride ions by the equivalent amount of bromide ions, and the work is consistently built accordingly.

Yet the very major issue to be resolved is that considering a 0.5 M concentration for the bromide ions would imply an accumulation of bromide salts that is one thousand times that in the original salt water. Although certain accumulation of bromide can be justified, it would be accordingly accompanied by the same-fold increase of chloride ions.

Shouldn't rather investigate a 5- or 10-fold increase of chloride ion concentration to be a more realistic scenario than the 1000-fold increase of bromide (without change in chloride ion concentration). Certainly a change in the pitting initiation mechanism should be expected just by increasing the chloride concentration.

(2) The Authors should consider more adequately the different mechanisms that are currently employed for describing pitting initiation. It is evident from the reading of the manuscript that the Authors have a preference for the adsorption model, but no adequate description in the Supplementary (lines 186-215, including Supp Fig 18) is given for the other two models. Even in Supp. Fig. 18 only one sketch (quite initial for B and late stage for C) is presented for those two models, which is not adequately describing them.

The same trend is followed in the body of the manuscript, when only the adsorption model is employed in lines 176-182. Such procedure is based on the observations from Raman spectra described in lines 172-175 as "thus verifying the adsorption-induced mechanism", and then present their observations "based on the above mechanism" (line 176).

The true thing is that the Raman data can "satisfy" the three models, and the most the authors may choose is modifying line 174 as to say "thus satisfying..." and indicate that they chose the adsorption-induced model as their preference for building the chemical reactions (1)-(4)

(2.1) In addition, it must be noticed that Oxygen is not balanced in equation (2)

(3) Further concern is about distinguishing three states sequentially affecting the oxide layer on nickel upon anodic polarization, namely, (1) stable passivation state, (2) metastable pitting state, and (3) pitting corrosion state. Indeed, these are 3 stages of the pitting nucleation mechanism, and they are affected by polarization as one to be favoured rather than the other. But they are associated with the intrinsic "instability of the passive oxide layer", for there is no such thing as a "stable passivation state". The authors are recommended to consider the work by Burstein as an introduction to the issue:

Burstein et al., Origins of pitting corrosion. *Corrosion Engineering, Science and Technology* 39 (2004) 25-30. <https://doi.org/10.1179/147842204225016859>.

Burstein and Pistorius, *Metastable Pitting Corrosion of Stainless Steel and the Transition to Stability* (1992) *Philosophical Transactions of the Royal Society A: Mathematical, Physical and Engineering Sciences*, 341 A, p. 531.

(4) The EIS data is not adequately presented:

4.1. Bode-amplitude graphs are missing. The choice for Nyquist plots offer the complete impedance data at each point although the frequency at which they were measured is missing in the graph. Therefore, Bode plots are often preferred as a more complete description of EIS data, but they should be Bode-amplitude and Bode-phase graphs (and not only one of them).

4.2. Following item 4.1., "Bode plots" is not an adequate description for the plots in the manuscript, but they are "Bode-phase plots" exclusively. For instance, see the legend in Fig 1 D,E

4.2. There is no single support to check the quality of the EIS analysis and adequacy of the equivalent circuits (EC) given in Supp. Fig 12 that are employed to extract the impedance parameters plotted in various graphs both in the body of the manuscript and in the supplementary information file. The authors should provide the simulated (fitted) spectra using those (EC) and in Supp Tables 3 and 4 by plotting solid lines in every impedance spectra given in Nyquist and Bode presentations.

4.3. The EC shown in Supp Fig 12B does not correspond with the sketch in the same figure. The sketch corresponds to processes occurring in parallel, but the EC is in a series arrangement.

4.4. Nyquist presentations are not adequately plotted, for the ticks are not equally separated in the X and Y axis although they correspond to the same units.

4.5. The R_p terms should not be named as "charge transfer resistances" but as "polarization resistances"

(5) What is a Tafel slope pit value? Tafel values have only meaning when the metal is not coated by an oxide, and they are usually measured around the corrosion potential. How can a Tafel analysis be employed for describing pitting of a breaking oxide layer? Show evidences in the literature. Show in the sketch in Supp Figure 1.

Further issues to be addressed:

(6) Figure legends should be more self-explanatory. Give operation measurement conditions in the legends, such as scan rates for CPCs and CVs, etc.

(7) How the surface area was determined for the nickel foam and mesh presentations? The CPCs in Fig. 1A are plotted in terms of current densities?

(8) How can some of the impedance data be normalized by the area but other not? Cf. Fig. 1D and 1F, Supp. Tables 3 and 4, Supp. Fig. 10, 11, 14C, 15.

(9) The SVET maps in Figure 2 show a huge unbalance between the total anodic and cathodic ionic currents. Comment on it and give the actual times for each map. Give experimental conditions.

In addition, give the time needed to complete one map, and what the correspondence is between "cycle" and time in Figure 2D E and line 327.

Give the tip-substrate distance.

Describe in lines 317-327 how sample polarization was achieved in the SVET measurements.

(10) Revise the word "gentle" in the context of stable pitting condition in line 234.

(11) It is nonsense to give concentrations of electrolytes in molar units invoking hydrated salts (line 285).

(12) The data in Supp Fig 6 might be consistent with SEM-EDX mappings.

(13) The measured data in Supp Fig. 7 are "experimental" instead of "original"

(14) Supp. Fig. 8 – Add arrows to indicate the scan direction in the different parts of the CPC and CV

(15) Suppl. Line 129 – Are you meaning a "threshold" potential for Ni²⁺/Ni³⁺ oxidations

(16) Give references to the interpretation of CPE-n values in Suppl. Lines 157-162.

Reviewer #3 (Remarks to the Author):

In this study, the authors revealed the Cl⁻-induced and Br⁻-induced pit corrosion processes of Ni-based anodes, which has certain significance for designing anti-Br-corrosion anodes in the future. However, further experimental and simulation results need to be discussed, such as the adsorption situation of different ions on Ni-based anodes. At the same time, the English in the text needs to be carefully considered. Therefore, rejection has to be suggested.

Response to Reviewer 1

General Comment: *Seawater electrolysis in alkaline environment is a reasonable and scalable pathway in seawater splitting field due to its low energy consumption, high selectivity and compatibility with current alkaline electrolyser. However, as the electrolyte should be reused, the accumulation of anions in electrolyte will definitely lead to severe electrode corrosion. It is well-known that Cl⁻ in seawater attacks the electrode, while the authors in this manuscript highlight that the accumulated Br⁻ in seawater is even more serious in terms of the corrosion behavior. Based on the detailed corrosion mechanism on conventional Ni substrate, it is quite convincing that Br⁻ corrosion should be taken into consideration in the future study of alkaline seawater splitting. This is an interesting study, which is of both fundamental and practical significance for the application of seawater splitting. Thus, I recommended this work to be accepted after minor revision.*

Response: We thank the reviewer for reviewing our manuscript and appreciate your precious comments and suggestions. We have revised our manuscript according to your suggestions.

Comment 1: *The authors conducted the experiments in 0.5 M NaCl and 0.5 M NaBr. Although the difference is obvious, it cannot directly reflect the corrosion of natural seawater, in which the concentration of Br⁻ is significantly lower than that of Cl⁻. Thus, I suggest the authors test the corrosion behavior of natural seawater on Ni substrate to clarify the role of trace amount of Br⁻.*

Response: Thanks for the valuable suggestion. The stability of Ni foil in the 1 M NaOH + seawater at a current density of 50 mA cm⁻² was obtained to investigate the corrosion behavior of Ni foil in the real seawater (Fig. R1). By the comparison of the corrosion behaviors in ClE and BrE, both narrow-deep (ClE) and wide-shallow (BrE) pits were found on Ni foil (Fig. R1d), indicating that the **Ni substrates exhibited a mixed corrosion behavior in alkaline natural seawater. Thus, the influence of trace Br⁻ (0.53 mM) on the corrosion is also significant.** Moreover, the corrosion behavior of typical NiFe-LDH anode in alkaline seawater was also conducted to demonstrate the role of trace Br⁻ (Fig. R2). After 30 h operation, both the narrow-deep pits (ClE) and the spalling phenomenon of the catalyst layer (BrE) can be observed on the substrate, further illustrating the non-negligible effect of trace Br⁻ on the corrosion of Ni-based catalysts. Overall, we analyzed the mixed corrosion behavior of Ni-based electrodes in alkaline-based natural seawater, distinguishing between

CIE and BrE (Note, the BrE is significant even with trace Br^-). The experimental results and corresponding discussions have been added in the Revised Manuscript and Supplementary Information (Fig. 5 and Supplementary Fig.22).

Fig. R1 **a** The LSV curves of Ni foil anode measured in 1 M NaOH + 0.5 M NaCl, 1 M NaOH + seawater, and 1 M NaOH + 0.5 M NaBr electrolytes, respectively. The scan rate was 5 mV s^{-1} . **b** The corresponding durability tests of Ni foil at a current density of 50 mA cm^{-2} . **c**, **d**, **e** Corresponding SEM images of Ni foil observed after 5 h operation in 1 M NaOH + 0.5 M NaCl, 1 h operation in 1 M NaOH + seawater, and 5 h operation in 1 M NaOH + 0.5 M NaBr, respectively.

Fig. R2 **a** The LSV curves of NiFe-E anode measured in 1 M NaOH + 0.5 M NaCl, 1 M NaOH + 0.5 M NaBr, and 1 M NaOH + seawater electrolytes, respectively. The scan rate was 5 mV s^{-1} . **b** The corresponding durability tests of NiFe-E at a current density of 400 mA cm^{-2} . **c, d, e** Corresponding SEM images of NiFe-E observed after 20 h operation in 1 M NaOH + 0.5 M NaCl, 20 h operation in 1 M NaOH + 0.5 M NaBr, and 30 h operation in 1 M NaOH + seawater, respectively.

Comment 2: *In this work, the corrosion behavior of NiFe-LDH nanoarrays electrode in Br⁻ and Cl⁻ has been carefully investigated. What about other catalytic materials?*

Response: To address your question, the typical OER electrocatalysts including NiCo-LDH nanoarrays/Ni foam (NiCo-LDH) and phosphorus-doped NiFe-LDH nanoarrays/Ni foam (NiFeP) were tested in 1 M NaOH + 0.5 M NaCl and 1 M NaOH + 0.5 M NaBr electrolytes to investigate the corrosion behavior for Cl⁻E and Br⁻E. The results (Fig. R3 and R4) revealed that the corrosion behaviors of NiCo-LDH and NiFeP in Br⁻ and Cl⁻ were comparable to that of the NiFe-E electrode, indicating that **Br⁻ corrosion induced spalling of the catalyst layer was a common phenomenon in integrated electrodes (substrate + catalyst on surface)**. The corresponding results have been added in the Revised Manuscript and Supplementary Information (Supplementary Fig.23-24).

Fig. R3 **a** The LSV curves of NiCo-LDH anode measured in 1 M NaOH + 0.5 M NaCl and 1 M NaOH + 0.5 M NaBr electrolytes. The scan rate was 5 mV s⁻¹. **b** The corresponding durability tests of NiCo-LDH at a current density of 200 mA cm⁻². **c**, **d**, **e** Corresponding SEM morphologies of original NiCo-LDH and morphologies observed after 24 h operation in 1 M NaOH + 0.5 M NaCl and in 1 M NaOH + 0.5 M NaBr.

Fig. R4 **a** The LSV curves of NiFeP anode measured in 1 M NaOH + 0.5 M NaCl and 1 M NaOH + 0.5 M NaBr electrolytes. The scan rate was 5 mV s^{-1} . **b** The corresponding durability tests of NiFeP at a current density of 400 mA cm^{-2} . **c, d, e** Corresponding SEM morphologies of original NiFeP and morphologies observed after 70 h operation in 1 M NaOH + 0.5 M NaCl and in 1 M NaOH + 0.5 M NaBr.

Comment 3: *I also want to know the different corrosion behaviors of the same catalyst in Cl^- and Br^- with varied morphologies.*

Response: To answer your question, we synthesized the NiFe-LDH nanoarrays electrode with different morphologies by electrodeposition (NiFe-E, Fig. R5a) and hydrothermal method (NiFe-H, Fig. R5d). Their corrosion behaviors were studied in 1 M NaOH + 0.5 M NaCl and 1 M NaOH + 0.5 M NaBr electrolytes. These results illustrated that both NiFe-E and NiFe-H exhibited similar corrosion behaviors in the same electrolyte, where Cl^- led to the formation of pits and Br^- caused the exfoliation of catalyst layer. The corresponding results have been added in the Revised Manuscript and Supplementary Information (Supplementary Fig.25).

Fig. R5 a, d SEM morphologies of NiFe-E and NiFe-H. **b, c** SEM morphologies of NiFe-E observed after 20 h operation in 1 M NaOH + 0.5 M NaCl and in 1 M NaOH + 0.5 M NaBr. **e, f** SEM morphologies of NiFe-H observed after 48 h operation in 1 M NaOH + 0.5 M NaCl and in 1 M NaOH + 0.5 M NaBr.

Comment 4: *At the beginning of this paper, the corrosion behavior of catalysts with the same concentration (0.5 M) of Cl^- and Br^- in 1 mM NaOH electrolyte was investigated. However, in Figure 5, the electrolyte used is 1 M NaOH. It's a little confusing. Is the conclusion that Br^- leads to more severe corrosion also applicable to seawater with higher alkali concentrations?*

Response: Thanks for your question, and we are sorry for causing any confusion. At the beginning of this paper, 1 mM NaOH electrolyte was used to study the corrosion behavior and corrosion mechanism. The choice of such low alkali concentration can avoid the signal interference of OER and $\text{Ni}^{2+}/\text{Ni}^{3+}$ redox reaction in the CPC test. If the pH of electrolyte is high, for example, ~ 14 , the E_{pit} would be covered by the $\text{Ni}^{2+}/\text{Ni}^{3+}$ redox peak and onset potential of OER (Fig. R6, see details in Supplementary Fig. 9), which was inconducive to the CPC assessments. In addition, the high pH would cause the rapid formation of the corrosion products ($\text{Ni}(\text{OH})_2$), resulting in coverage of the pits. While in Fig. 5b, for the purposes of approaching practical operation condition, we used 1 M NaOH to investigate the corrosion behaviors of NiFe-E electrodes. The conclusion is that Br^- was also more aggressive than Cl^- even in high alkali concentration. We apologize again that changing the pH of electrolyte in

Fig. 5b is indeed a little confusing. Thus, we have revised this part to clarify our claims in the Revised Manuscript.

Fig. R6 a The CPC of Ni foam measured in 1 M NaOH + 0.5 M NaCl electrolyte, and the scan rate was 10 mV s^{-1} . **b** The corresponding Cyclic voltammetry (CV) curve, and the scan rate was 10 mV s^{-1} .

Comment 5: *How did the authors determine computational model? Any evidence from experiments?*

Response: Thanks for the question. According to the previous studies^{1,2}, metal Ni substrate was covered by a spontaneous generated passive film (NiO), which was consistent with our SEM-EDX mappings and XPS results (Supplementary Fig. 6-8). Moreover, as the positive potential was applied, NiO would also form on the surface of Ni substrate spontaneously.^{3,4} As for pitting corrosion, the pit initiated at the outside of passive film (i.e., NiO), where halide adsorbed and then corroded the passive film.^{5,6} Thus, we constructed a nickel oxide model to study the pitting mechanism. Typically, there are three widely accepted mechanisms for pitting initiation, including (1) adsorption-induced mechanism, (2) ion migration and penetration model, and (3) mechanical film breakdown theory (Fig. R7).^{1,4-11} Note, the mechanical film breakdown theory was excluded here because it was mainly caused by mechanical stress. In our case, we considered the pitting initiation of Ni substrates in ClE and BrE followed a mixture of both adsorption and permeation mechanisms. Therefore, we conducted the density functional theory simulations to study the adsorption-induced mechanism and carried out the nudged elastic band calculations to investigate the ion migration and penetration model. The additional discussions were added in the Revised Manuscript and Supplementary Information.

Fig. R7 a, b, c Schematic illustrations of three mechanisms for pit initiation, including adsorption-induced mechanism, ion migration and penetration model, and mechanical film breakdown theory.

Comment 6: It is suggested that the corrosion processes of Ni under the electrolyte with both Cl^- and Br^- should be illustrated. Please demonstrate it in Fig. 4.

Response: According to the suggestion, we have studied the corrosion process of Ni foil in 1 mM NaOH + 0.25 M NaCl + 0.25 M NaBr electrolyte (ClBrE) during the potentiostatic polarization test at a constant potential of 1.33 V vs. RHE. As shown in Fig. R8a, the current of Ni foil in ClBrE increased fast and then gradually. The corresponding *ex-situ* SEM images showed that Br^- would corrode the Ni foil first, and then Cl^- would continue to corrode at the Br^- -induced pits (Fig. R8b). Based on the experimental results, we supplemented the schematic diagram of corrosion process of Ni under the electrolyte with both Cl^- and Br^- (Fig. R9), which has been added in the Revised Manuscript (Fig. 2 and Fig. 4).

Fig. R8 a Potentiostatic polarization curves of Ni foil tested in CIE, BrE, and ClBrE, respectively. **b** Corresponding morphology of pits on Ni foil corroded by Cl^- , Br^- , and both Cl^- and Br^- during the potentiostatic polarization tests.

Fig. R9 a, b, c Schematic illustrations of Ni foil corroded by Cl⁻, Br⁻, and both Cl⁻ and Br⁻, respectively. Insets are detailed views of NiO passive film corroded by halide.

References:

- 1 Sharma, S. K. *Green corrosion chemistry and engineering* (Wiley - VCH Verlag GmbH & Co. KGaA, 2011).
- 2 Cramer, S. D. & Covino, B. S. *ASM Handbook: v. 13A Corrosion: Fundamentals, Testing, and Protection* (ASM International, 2003).
- 3 Grdeń, M. & Klimek, K. EQCM studies on oxidation of metallic nickel electrode in basic solutions. *J. Electroanal. Chem.* **581**, 122-131 (2005).
- 4 Hoar, T. P. The production and breakdown of the passivity of metals. *Corros. Sci.* **7**, 341-355 (1967).
- 5 Kolotyrkin, J. M. Effects of anions on the dissolution kinetics of metals. *J. Electrochem. Soc.* **108**, 209 (1961).

- 6 Soltis, J. Passivity breakdown, pit initiation and propagation of pits in metallic materials - review. *Corros. Sci.* **90**, 5-22 (2015).
 - 7 Frankel, G. S. Pitting corrosion of metals: a review of the critical factors. *J. Electrochem. Soc.* **145**, 2186-2198 (1998).
 - 8 Zhang, B. et al. Unmasking chloride attack on the passive film of metals. *Nat. Commun.* **9**, 2559 (2018).
 - 9 Marcus, P., Maurice, V. & Strehblow, H. H. Localized corrosion (pitting): a model of passivity breakdown including the role of the oxide layer nanostructure. *Corros. Sci.* **50**, 2698-2704 (2008).
 - 10 Schmuki, P. From bacon to barriers: a review on the passivity of metals and alloys. *J. Solid State Electrochem.* **6**, 145-164 (2014).
 - 11 Burstein, G. T., Liu, C., Souto, R. M. & Vines, S. P. Origins of pitting corrosion. *Corros. Eng. Sci. Techn.* **39**, 25-30 (2013).
-

Response to Reviewer 2

General Comment: *The work under consideration maybe regarded of significance to the fields of corrosion and energy generation using electrochemical methods. In particular, the enhanced corrosion rates experienced by electrodes in marine environment compared to simulated laboratory tests is certainly a hot topic. But the following issues need addressing:*

Response: We sincerely appreciate the insightful comments and constructive suggestions from the reviewer, which help us significantly improve the quality of our manuscript. We have revised our manuscript, and the details are further discussed in the following point-by-point response.

Comment 1: *The Authors in this work claim the neglected contribution of bromide ions to be the main source for such discrepancy, and they justify such effect due to the increased amount of salts at the electrode compartment due to the operation of the electrochemical cell as the rationale for investigating the effect of bromide ions.*

Accordingly, a study is presented by replacing the typical concentration of chloride ions by the equivalent amount of bromide ions, and the work is consistently built accordingly.

Yet the very major issue to be resolved is that considering a 0.5 M concentration for the bromide ions would imply an accumulation of bromide salts that is one thousand times that in the original salt water. Although certain accumulation of bromide can be justified, it would be accordingly accompanied by the same-fold increase of chloride ions.

Shouldn't rather investigate a 5- or 10-fold increase of chloride ion concentration to be a more realistic scenario than the 1000-fold increase of bromide (without change in chloride ion concentration). Certainly a change in the pitting initiation mechanism should be expected just by increasing the chloride concentration.

Response: First of all, the comments are very useful to our manuscript. Indeed, in the practical operation conditions, with the replenishment of seawater, NaCl will definitely accumulate to the saturated concentration (~5 M), which is fatal to the electrodes. For this reason, our team has reported that the increase of NaOH concentration can effectively reduce the saturated NaCl content in the electrolyte.¹ Due to the common-ion effect, the solubility of NaCl will significantly decrease from ~5 M to ~2.5 M when the concentration of NaOH increases to 6 M. As the decrease of Cl⁻ content and the corrosion-inhibiting effect of OH⁻, the

corrosion of electrode can be alleviated, which is conducive to the stability of the anode. Thus, electrodes operating in a 6M NaOH + seawater electrolyte must be the future trend of practical application.

Notably, the concentration of Br^- will also increase with the continuous addition of seawater. More importantly, as the solubility of NaBr is higher than NaCl in water², it will cause the decrease of saturation solubility of NaCl in electrolytes containing 6M NaOH. Therefore, with the reaction time prolonging, the concentration of Cl^- will be less than 2.5 M in 6M NaOH electrolyte, while **the concentration of Br^- will continuously increase** (Fig. R10). **It is predictable that after some time, Br-induced corrosion of anodes will dominate.**

Fig. R10 Schematic diagram of the accumulation of halides in the electrolyte with the replenishment of seawater during seawater electrolysis process.

Herein, according to your suggestion, we found that Br^- had a non-negligible influence on the corrosion behavior of anode even if the Cl^- concentration was increased by 5-fold. Here we firstly chose 1 mM NaOH + 2.5 M NaCl and 1 mM NaOH + 2 M NaCl + 0.5 M NaBr electrolytes to investigate the electrochemical plots of NiFe-LDH nanoarrays/Ni foam (synthesized by hydrothermal method, NiFe-H) anode. As shown in Fig. R11a-c, NiFe-H possessed a higher current in 1 mM NaOH + 2 M NaCl + 0.5 M NaBr electrolytes at the same potential ($>E_{\text{pit}}$). Furthermore, we also investigated the corrosion behaviors of NiFe-H anodes during the oxygen evolution reaction durability test in 1 M NaOH + 2.5 M NaCl and 1 M NaOH + 2 M NaCl + 0.5 M NaBr electrolytes at the current density of 400 mA cm^{-2} . Here we increased the alkali concentration to approach the practical operation condition, and the reason for not using 6 M NaOH was that the corrosion rate of the anode was too slow and time-consuming at such a high pH. The lifetime of NiFe-H anode was drastically reduced in 1

M NaOH + 2 M NaCl + 0.5 M NaBr electrolyte (Fig. R12b), indicating the harmful corrosion effect of Br⁻. The morphologies of NiFe-H anodes after the 2 h durability test in these two electrolytes were observed. In 1 M NaOH + 2.5 M NaCl electrolyte, only some pits appeared on the surface of anode (Fig. R12c), while in 1 M NaOH + 2 M NaCl + 0.5 M NaBr electrolyte, the spalling of catalyst layer accompanied by some pits on the Ni substrate can be observed (Fig. R12d).

Based on the above results, we can conclude that **it is necessary to investigate the effect of Br⁻ on the corrosion of anodes, and even at high salt concentrations, the corrosion behaviors of anode in chloride and bromide are similar to those of low salt concentrations.** The corresponding results have been added in the Revised Manuscript and Supplementary Information (Supplementary Fig. 3 and 26).

Fig. R11 **a** CV curves of NiFe-H measured in 1 mM NaOH + 2.5 M NaCl and 1 mM NaOH + 2 M NaCl + 0.5 M NaBr electrolytes, respectively. The scan rates were 10 mV s⁻¹. **b** Corresponding CVCs of NiFe-H. The scan rates were 10 mV s⁻¹. **c** Potentiostatic polarization curves of NiFe-H tested in 1 mM NaOH + 2.5 M NaCl and 1 mM NaOH + 2 M NaCl + 0.5 M NaBr electrolytes.

Fig. R12 **a** LSV curves of NiFe-H measured in 1 M NaOH + 2.5 M NaCl and 1 M NaOH + 2 M NaCl + 0.5 M NaBr electrolytes, respectively. The scan rates were 5 mV s⁻¹. **b** The corresponding durability of NiFe-H at a current density of 400 mA cm⁻². **c, d** Corresponding morphology of NiFe-H observed after 2 h operations.

Comment 2.1: *The Authors should consider more adequately the different mechanisms that are currently employed for describing pitting initiation. It is evident from the reading of the manuscript that the Authors have a preference for the adsorption model, but no adequate description in the Supplementary (lines 186-215, including Supp Fig 18) is given for the other two models. Even in Supp. Fig. 18 only one sketch (quite initial for B and late stage for C) is presented for those two models, which is not adequately describing them.*

The same trend is followed in the body of the manuscript, when only the adsorption model is employed in lines 176-182. Such procedure is based on the observations from Raman spectra described in lines 172-175 as "thus verifying the adsorption-induced mechanism", and then present their observations "based on the above mechanism" (line 176).

The true thing is that the Raman data can "satisfy" the three models, and the most the authors may choose is modifying line 174 as to say "thus satisfying..." and indicate that they chose the adsorption-induced model as their preference for building the chemical reactions (1)-(4)

Response: Thanks for your suggestion, according to this, we have carefully scrutinized the previous studies on pitting initiation mechanisms and supplemented the penetration model (Fig. R13b) and film breakdown theory (Fig. R13c).³⁻¹¹ A detailed description of these mechanisms is as follows.

(a) Adsorption-induced mechanism. When the voltage is applied to anodes, halide ions will adsorb on the anode surface and then replace the surface oxygen of the passive layer, leading to the formation of a complex with metal ions. Then the complex will dissolve and diffuse into the solution to form the hydroxide. Specifically, the passivation rate is faster than corrosion rate ($i_{\text{passivation}} > i_{\text{pitting}}$) at the beginning, the passive layer becomes thicker. When the corrosion rate increases and becomes larger than passivation rate ($i_{\text{pitting}} > i_{\text{passivation}}$), the passive layer will become thinner at the corrosion sites. As the corrosion rate continues to increase and becomes much higher than passivation rate ($i_{\text{pitting}} \gg i_{\text{passivation}}$), the breakage of passive film and the corrosion of the underneath substrate will happen.

(b) Ion migration and penetration model. Due to the influence of the high electric field, halide ions will penetrate and contaminate the oxide film, causing higher electrical and ionic conductivity along the penetration paths. During the penetration process, some vacancies may also form in the passive film. Thereby the rapid release of cation at the film-solution interface or the accumulation of vacancies at the metal-film interface may happen. Finally, the passive film will break down and followed by the corrosion of substrate.

(c) Mechanical film breakdown theory. Some mechanical factors, including electrostriction stress, blistering, and micro-capillary formation, may lead to the strain on the passive film. At some weak sites, the stress on the film may exceed the mechanical breakdown stress, causing the formation of cracks in the film. Then the film will rupture rapidly and form a channel where the halide ions can pass through to corrode the underneath metal.

Note that the mechanical film breakdown usually caused by mechanical factors, which was less relevant to the halide species. Thus, we consider that the pitting initiation of Ni substrates in ClE and BrE mainly involves a combination of adsorption and permeation mechanisms. Therefore, we conducted the density functional theory simulations to study the adsorption and carried out the nudged elastic band calculations to investigate the diffusion.

As for Raman data, we agree with your point that Raman data can satisfy the three models. The discussion “thus verifying the adsorption-induced mechanism” was indeed not reasonable. Since we believed our case followed the combination mechanisms, we decided that it would be more appropriate to remove the discussion “and thus verifying the adsorption-induced

mechanism”. The above contents have been added in the Revised Manuscript and Supplementary Information (Supplementary Fig.16).

Fig. R13 a, b, c Schematic illustrations of three mechanisms for pit initiation, including adsorption-induced mechanism, ion migration and penetration model, and mechanical film breakdown theory.

Comment 2.2: *In addition, it must be noticed that Oxygen is not balanced in equation (2)*

Response: Thanks for the comment. We double-checked the number of oxygen atoms in equation (2), and found that the Oxygen was balanced.

The number of oxygen atoms on the left side of the equation was $1+y$, which was equal to the right side $1-y+2y=1+y$.

Comment 3: *Further concern is about distinguishing three states sequentially affecting the oxide layer on nickel upon anodic polarization, namely, (1) stable passivation state, (2) metastable pitting state, and (3) pitting corrosion state. Indeed, these are 3 stages of the pitting nucleation mechanism, and they are affected by polarization as one to be favoured rather than the other.*

But they are associated with the intrinsic “instability of the passive oxide layer”, for there is no such thing as a “stable passivation state”. The authors are recommended to consider the work by Burstein as an introduction to the issue:

Burstein et al., Origins of pitting corrosion. Corrosion Engineering, Science and Technology 39 (2004) 25-30. <https://doi.org/10.1179/147842204225016859>.

Burstein and Pistorius, Metastable Pitting Corrosion of Stainless Steel and the Transition to Stability (1992) Philosophical Transactions of the Royal Society A: Mathematical, Physical and Engineering Sciences, 341 A, p. 531.

Response: Thanks for pointing out the mistakes made by us. We have carefully read these two articles about the nucleation, metastable growth, and stable growth of pits and sorted out these concepts. We misused the terms that describe the three states in the CPC (Supplementary Fig. 3 in the previous manuscript) in the pitting nucleation mechanism sketches (Fig. 4a in the previous manuscript). We have removed these three terms described in the pitting nucleation mechanism (see details in the comments 2.1) and cited these two articles in our manuscript. Furthermore, we replaced the previously described three states in the CPC with more appropriate terms, including passive dominant state, metastable pitting corrosion state, and stable pit growth state. The revisions were highlighted in the Revised Manuscript and Supplementary Information.

Comment 4.1: *The EIS data is not adequately presented: Bode-amplitude graphs are missing. The choice for Nyquist plots offer the complete impedance data at each point although the frequency at which they were measured is missing in the graph. Therefore, Bode plots are often preferred as a more complete description of EIS data, but they should be Bode-amplitude and Bode-phase graphs (and not only one of them). 4.2. Following item 4.1., “Bode plots” is not an adequate description for the plots in the manuscript, but they are “Bode-phase plots” exclusively. For instance, see the legend in Fig 1 D,E.*

Response: Thanks for correcting our mistakes. We have supplemented the Bode plots with both Bode-phase and Bode-amplitude graphs (Fig. R14-15), and we have corrected the legend “Bode-phase plots” in Fig. 1 d, e, which was highlighted in the Revised Manuscript.

Comment 4.2: *There is no single support to check the quality of the EIS analysis and adequacy of the equivalent circuits (EC) given in Supp. Fig 12 that are employed to extract the impedance parameters plotted in various graphs both in the body of the manuscript and in the supplementary information file. The authors should provide the simulated (fitted) spectra using those (EC) and in Supp Tables 3 and 4 by plotting solid lines in every impedance spectra given in Nyquist and Bode presentations.*

Response: We have added the fitted data to the Bode and Nyquist plot (Fig. R14); however, the fitting curves of high potential did not match very well at low frequencies. As shown in

Fig. R14 c and f, with the occurrence of corrosion, the low-frequency inductive behavior could be observed in the Nyquist plots, which may be attributed to the metal dissolution process^{12,13} or the adsorption of intermediate species.¹⁴ Besides, the corrosion process also contained the diffusion of reactants and products, increasing the complexity of the impedance analysis. We have tried a variety of ECs to fit the impedance spectra. Unfortunately, none of them fitted the low-frequency part of impedance spectra pretty well. The good thing is that ECs used in our manuscript were well fitted in the mid-high frequency region of impedance spectra. This region corresponded to the $\text{Ni}^0 \rightarrow \text{Ni}^{2+}$ transition, denoting the corrosion process. Thus, we believed it is reasonable to use such simple ECs to roughly assess the corrosion process and draw the conclusion that Ni foil was easier to be corroded by Br^- than Cl^- .

Fig. R14 a, b, d, e Bode-phase and Bode-amplitude plots of Ni foil at different potentials in CIE and BrE, respectively. **c, f** Corresponding Nyquist plots of Ni foil at different potentials in CIE and BrE, respectively.

As for the time-dependent Bode plots (Fig. R15), we used the same EC as the potential-dependent Bode plots (Fig. R14) to quantify the CPE-n to investigate the change in the surface morphology of anode during the corrosion process. However, the fitting curves also did not match very well in the low-frequency region. **To obtain a more accurate value of CPE-n, the graphical method^{15,16} was adopted. We plotted the imaginary part of the impedance-frequency plots, where the slope at high frequency was $-a$ (i.e., CPE-n).**

Therefore, by fitting the slope in the linear part of the high-frequency region, we obtained the variation of α with cycles, and the results were shown in Fig. R16. As the number of cycles increased, α of Ni foil in CIE decreased gradually to less than 0.5, suggesting the formation of severe pitting pores. In contrast, α of Ni foil in BrE were maintained, indicating that the electrode surface became rough due to corrosion.^{17,18} The corresponding results have been added in the Revised Manuscript and Supplementary Information (Fig. 2e and Supplementary Fig.15).

Fig. R15 a, b, d, e Time-dependent Bode-phase and Bode-amplitude plots of Ni foil in CIE and BrE, respectively. c, f Corresponding imaginary part of the impedance-frequency plots.

Fig. R16 α -cycle plots obtained by fitting the slope of linear part in the high frequency region in time-dependent imaginary part of the impedance-frequency plots.

Comment 4.3: The EC shown in Supp Fig 12B does not correspond with the sketch in the same figure. The sketch corresponds to processes occurring in parallel, but the EC is in a series arrangement.

Response: Thanks for pointing out our mistakes. The sketch was wrong because NiOOH was transformed from Ni²⁺, which was generated on the outermost surface of the anode. We have rectified the sketch (Fig. R17b) and highlighted it in Supplementary Information.

Fig. R17 a, b Equivalent circuit models for pitting corrosion of Ni foil in different potential intervals, for 1.03 V to 1.38 V, for 1.43 V to 1.48 V.

Comment 4.4: Nyquist presentations are not adequately plotted, for the ticks are not equally separated in the X and Y axis although they correspond to the same units.

Response: We do appreciate you pointing out our mistakes. We have corrected the Nyquist presentations (Fig. R14c, f).

Comment 4.5: The R_p terms should not be named as “charge transfer resistances” but as “polarization resistances”.

Response: Thanks for your suggestion. We have changed the explanation of R_p and highlighted it in the Revised Manuscript.

“where R_{p1} and R_{p2} are the polarization resistances at different frequency regions.”

Comment 5: What is a Tafel slope_{pit} value? Tafel values have only meaning when the metal is not coated by an oxide, and they are usually measured around the corrosion potential. How can a Tafel analysis be employed for describing pitting of a breaking oxide layer? Show evidences in the literature. Show in the sketch in Supp Figure 1.

Response: Thanks for your question. Tafel slope_{pit} in the manuscript denotes the linear relationship between logarithm of current density and the potential in the region where the potential was larger than E_{pit} (Fig. R18). In electrocatalysis reactions, the Tafel slope,

typically measured near the onset potential, is usually employed to evaluate the reaction rate.¹⁹⁻²¹ Analogously, we believe that Tafel slope_{pit} can evaluate the corrosion reaction rate during the stable growth of pits. The corresponding revisions have been added in the Supplementary Information (Supplementary Fig. 4).

Fig. R18 Schematic representation of a generic anodic CPC.

Further issues to be addressed:

Comment 6: *Figure legends should be more self-explanatory. Give operation measurement conditions in the legends, such as scan rates for CPCs and CVs, etc.*

Response: Thanks for your suggestion. We have added the operation measurement conditions in the legends, which were highlighted in the Revised Manuscript.

“CPCs of Ni substrates tested in CIE and BrE at a scan rate of 10 mV s^{-1} .”

“The corresponding CV curves of Ni foil, Ni foam, and Ni mesh measured in CIE and BrE, and the scan rate was 1 mV s^{-1} .”

Comment 7: *How the surface area was determined for the nickel foam and mesh presentations? The CPCs in Fig. 1A are plotted in terms of current densities?*

Response: Thanks for your question. It was our negligence that we did not clarify the area of the nickel foam and mesh as a geometrical area rather than a surface area. We have changed the current density “ j ” to “ j_{geo} ” in Fig. 1a.

Comment 8: *How can some of the impedance data be normalized by the area but other not? Cf. Fig. 1D and 1F, Supp. Tables 3 and 4, Supp. Fig. 10, 11, 14C, 15.*

Response: Thanks for the question. It was our mistake that “ cm^{-2} ” was omitted from the unit of $-Z'$. Actually, all of the impedance data were normalized by the area. The corresponding

figures (Fig. R14 and R15) have been revised in the Supplementary Information (Supplementary Fig. 15).

Comment 9: *The SVET maps in Figure 2 show a huge unbalance between the total anodic and cathodic ionic currents. Comment on it and give the actual times for each map. Give experimental conditions.*

In addition, give the time needed to complete one map, and what the correspondence is between “cycle” and time in Figure 2D E and line 327.

Give the tip-substrate distance.

Describe in lines 317-327 how sample polarization was achieved in the SVET measurements.

Response: Thanks for your suggestion. Theoretically, the anodic current should be equal to the cathodic current in the SVET maps. However, our SVET maps showed a huge unbalance between the total anodic and cathodic ionic currents. After reviewing the relevant literatures, we summarized two main reasons that may cause the unbalance. 1) The scan area was 4 mm × 4 mm, while the area of sample subjected to potentiostatic polarization was 1 cm × 2 cm. The total anodic and cathodic ionic currents were balanced over the entire sample, while they might be unbalanced in the local area (i.e., scan area). 2) Due to the time required to complete a map, the potential signals obtained were not simultaneous. The current distribution on the scan area might change during the scanning²², resulting in the huge unbalance.

Completing one map took 222 s, including a 5 s delay time and a 1 s restitution time. The time interval between each circle was 2 s, during which the data file was saved and named. The actual time for each map was 0 s, 224 s, and 448 s, which were added in Fig. 2c, d in the manuscript.

The tip-substrate distance was less than 100 μm.

There were two current circuits in the test, one was the circuit for potentiostatic polarization tests applied on the sample, and the other was the circuit for SVET measurements (Fig. R19). The potentiostatic polarization test was continually operated with an applied voltage of 1.3 V vs. RHE. The SVET measurements were conducted simultaneously with the potentiostatic polarization test.

The aforementioned contents have been added and highlighted in the Revised Manuscript and Supplementary Information.

Fig. R19 Schematic diagram of SVET measurements.

Comment 10: *Revise the word “gentle” in the context of stable pitting condition in line 234.*

Response: Thanks for your suggestion. The word “gentle” does not fit here, it might be better to change it to “slow”. But we have removed this sentence due to the change of Fig. 5.

Comment 11: *It is nonsense to give concentrations of electrolytes in molar units invoking hydrated salts (line 285).*

Response: Thank you for pointing out our mistakes. We have removed the crystalline water behind the salt in the Revised Manuscript.

“0.5 mmol of $\text{Ni}(\text{NO}_3)_2$, 0.5 mmol of $\text{Fe}(\text{NO}_3)_3$, and 10 mmol of urea, were dissolved in 35 mL of deionized water.”

Comment 12: *The data in Supp Fig 6 might be consistent with SEM-EDX mappings.*

Response: Thanks for the suggestion. We have revised “SEM mappings” to “SEM-EDX mappings” in the Supplementary Information.

“The results demonstrated that the Ni substrates were composed of Ni and O, which was consistent with the SEM-EDX mappings.”

Comment 13: *The measured data in Supp Fig. 7 are “experimental” instead of “original”.*

Response: Thanks for your suggestion. We have replaced the “experimental” with “original” in the figure (Fig. R20), which was highlight in the Supplementary Information.

Fig. R20 a, b, c High-resolution Ni 2p spectrum of Ni foil, Ni mesh, and Ni foam. The deconvoluted spectrums illustrated that the coexistence of Ni and NiO.

Comment 14: *Supp. Fig. 8 – Add arrows to indicate the scan direction in the different parts of the CPC and CV.*

Response: Thanks for your suggestion. We have added arrows in the CPC and CV in Supplementary Fig. 8, and the revised figure (Fig. R21) was highlighted in the Supplementary Information.

Fig. R21 a The CPC of Ni foam measured in 1 M NaOH + 0.5 M NaCl electrolyte, and the scan rate was 10 mV s⁻¹. **b** The corresponding Cyclic voltammetry (CV) curve, and the scan rate was 10 mV s⁻¹.

Comment 15: *Suppl. Line 129 – Are you meaning a “threshold” potential for Ni²⁺/Ni³⁺ oxidations.*

Response: Thanks for the question. Yes, the content in brackets in Suppl. Line 129 means the threshold potential for Ni²⁺/Ni³⁺ oxidations. We have revised “baseline” to “threshold” in the Supplementary Information.

“(the threshold for Ni²⁺/Ni³⁺ oxidation)”

Comment 16: *Give references to the interpretation of CPE-n values in Suppl. Lines 157-162.*

Response: Thanks for your suggestion. We have added the references to interpret the α (i.e., CPE-n) values, which was highlighted in the Supplementary Information.

References:

- 1 Li, P. et al. Common-ion effect triggered highly sustained seawater electrolysis with additional NaCl production. *Research* **2020**, 2872141 (2020).
 - 2 Pinho, S. P. & Macedo, E. A. Solubility of NaCl, NaBr, and KCl in water, methanol, ethanol, and their mixed solvents. *J. Chem. Eng. Data* **50**, 29-32 (2004).
 - 3 Sharma, S. K. *Green corrosion chemistry and engineering* (Wiley - VCH Verlag GmbH & Co. KGaA, 2011).
 - 4 Frankel, G. S. Pitting corrosion of metals: a review of the critical factors. *J. Electrochem. Soc.* **145**, 2186-2198 (1998).
 - 5 Soltis, J. Passivity breakdown, pit initiation and propagation of pits in metallic materials - review. *Corros. Sci.* **90**, 5-22 (2015).
 - 6 Kolotyrkin, J. M. Effects of anions on the dissolution kinetics of metals. *J. Electrochem. Soc.* **108**, 209 (1961).
 - 7 Zhang, B. et al. Unmasking chloride attack on the passive film of metals. *Nat. Commun.* **9**, 2559 (2018).
 - 8 Marcus, P., Maurice, V. & Strehblow, H. H. Localized corrosion (pitting): a model of passivity breakdown including the role of the oxide layer nanostructure. *Corros. Sci.* **50**, 2698-2704 (2008).
 - 9 Hoar, T. P. The production and breakdown of the passivity of metals. *Corros. Sci.* **7**, 341-355 (1967).
 - 10 Schmuki, P. From bacon to barriers: a review on the passivity of metals and alloys. *J. Solid State Electrochem.* **6**, 145-164 (2014).
 - 11 Burstein, G. T., Liu, C., Souto, R. M. & Vines, S. P. Origins of pitting corrosion. *Corros. Eng. Sci. Techn.* **39**, 25-30 (2013).
 - 12 Keddah, M., Mottos, O. R. & Takenouti, H. Reaction model for iron dissolution studied by electrode impedance: I . Experimental results and reaction model. *J. Electrochem. Soc.* **128**, 257-266 (1981).
 - 13 Epelboin, I. & Keddah, M. Faradaic impedances: Diffusion impedance and reaction impedance. *J. Electrochem. Soc.* **117**, 1052 (1970).
-

- 14 Mamlouk, M. & Scott, K. Analysis of high temperature polymer electrolyte membrane fuel cell electrodes using electrochemical impedance spectroscopy. *Electrochim. Acta* **56**, 5493-5512 (2011).
 - 15 Orazem, M. E., Pébère, N. & Tribollet, B. Enhanced graphical representation of electrochemical impedance data. *J. Electrochem. Soc.* **153**, B129 (2006).
 - 16 Jorcin, J., Orazem, M. E., Pébère, N. & Tribollet, B. CPE analysis by local electrochemical impedance spectroscopy. *Electrochim. Acta* **51**, 1473-1479 (2006).
 - 17 Vivier, V. & Orazem, M. E. Impedance analysis of electrochemical systems. *Chem. Rev.* **122**, 11131-11168 (2022).
 - 18 Amor, Y. B., Sutter, E. M. M., Takenouti, H., Orazem, M. E. & Tribollet, B. Interpretation of electrochemical impedance for corrosion of a coated silver film in terms of a pore-in-pore model. *J. Electrochem. Soc.* **161**, C573-C579 (2014).
 - 19 van der Heijden, O., Park, S., Eggebeen, J. J. J. & Koper, M. T. M. Non-kinetic effects convolute activity and tafel analysis for the alkaline oxygen evolution reaction on NiFeOOH electrocatalysts. *Angew. Chem. Int. Ed. Engl.* **62**, e202216477 (2023).
 - 20 Yang, H., Zhang, Y., Hu, F. & Wang, Q. Urchin-like CoP nanocrystals as hydrogen evolution reaction and oxygen reduction reaction dual-electrocatalyst with superior stability. *Nano Lett.* **15**, 7616-7620 (2015).
 - 21 Cao, D. et al. Engineering the in-plane structure of metallic phase molybdenum disulfide via Co and O dopants toward efficient alkaline hydrogen evolution. *ACS Nano* **13**, 11733-11740 (2019).
 - 22 Yan, M. et al. SVET method for characterizing anti-corrosion performance of metal-rich coatings. *Corros. Sci.* **52**, 2636-2642 (2010).
-

Response to Reviewer 3

General Comment: *In this study, the authors revealed the Cl⁻-induced and Br⁻-induced pit corrosion processes of Ni-based anodes, which has certain significance for designing anti-Br corrosion anodes in the future. However, further experimental and simulation results need to be discussed, such as the adsorption situation of different ions on Ni-based anodes. At the same time, the English in the text needs to be carefully considered. Therefore, rejection has to be suggested.*

Response: We thank this reviewer's comments on the importance of our work in revealing the corrosion mechanism of Ni-based anodes in halide.

Based on the suggestions from reviewer 1 and 2, we have made great efforts in further experimental supplements, mechanism analysis, data processing and analysis, mistakes correction, and English revision, to finally improve the quality of this manuscript. The improvements were shown as follows.

1. Further experimental supplements

Regarding the experiment aspects, we supplemented the investigation of corrosion behaviors of anodes in seawater and electrolytes with high salt concentrations. In addition, the corrosion behaviors of anodes with various catalyst layers in ClE and BrE were also studied to demonstrate general applicability of Br⁻-induced corrosion behavior.

1.1 Electrolytes

Firstly, we investigated the corrosion behaviors of Ni foil and NiFe-LDH synthesized by electrodeposition (NiFe-E) electrode in seawater, and found that both **Ni foil and NiFe-E electrode exhibited a mixed corrosion behavior in natural seawater**. As for Ni foil, compared with the corrosion behaviors in ClE and BrE, both wide-shallow and narrow-deep pits were formed on the Ni foil (Fig. R22d). For NiFe-E electrode, after 30 h operation in 1 M NaOH + seawater, both the narrow-deep pits (ClE) and the spalling phenomenon of the catalyst layer (BrE) can be observed on the substrate (Fig. R23d), further illustrating **the non-negligible effect of trace Br⁻ on the corrosion of Ni-based catalysts**. Thus, even though **Br⁻ is of trace amount (0.53 mM) in seawater, its influence on the corrosion behavior of Ni substrates should be attracted more attention**.

Fig. R22 **a** The LSV curves of Ni foil anode measured in 1 M NaOH + 0.5 M NaCl, 1 M NaOH + seawater, and 1 M NaOH + 0.5 M NaBr electrolytes, respectively. The scan rate was 5 mV s^{-1} . **b** The corresponding durability tests of Ni foil at a current density of 50 mA cm^{-2} . **c, d, e** Corresponding SEM images of Ni foil observed after 5 h operation in 1 M NaOH + 0.5 M NaCl, 1 h operation in 1 M NaOH + seawater, and 5 h operation in 1 M NaOH + 0.5 M NaBr, respectively.

Fig. R23 **a** The cyclic voltammograms of NiFe-E measured in 1 M NaOH + 0.5 M NaCl (blue) and 1 M NaOH + 0.5 M NaBr (red) electrolytes. The scan rate was 5 mV s^{-1} . **b** The corresponding durability tests of NiFe-E at a current density of 50 mA cm^{-2} . **c, d, e** Corresponding SEM images of NiFe-E observed after 20 h operation in 1 M NaOH + 0.5 M NaCl, 20 h operation in 1 M NaOH + 0.5 M NaBr, and 30 h operation in 1 M NaOH + seawater, respectively.

Fig. R23 a The LSV curves of NiFe-E anode measured in 1 M NaOH + 0.5 M NaCl, 1 M NaOH + 0.5 M NaBr, and 1 M NaOH + seawater electrolytes, respectively. The scan rate was 5 mV s⁻¹. **b** The corresponding durability tests of NiFe-E at a current density of 400 mA cm⁻². **c, d, e** Corresponding SEM images of NiFe-E observed after 20 h operation in 1 M NaOH + 0.5 M NaCl, 20 h operation in 1 M NaOH + 0.5 M NaBr, and 30 h operation in 1 M NaOH + seawater, respectively.

We also studied the corrosion processes of Ni in the electrolyte with both Cl⁻ and Br⁻ (ClBrE). As shown in Fig. R24a, the current of Ni foil in ClBrE increased fast and then gradually. The corresponding *ex-situ* SEM images showed that **Br⁻ would corrode the Ni foil first, and then Cl⁻ would continue to corrode at the Br⁻-induced pits** (Fig. R24b). Based on the experimental results, we supplemented the schematic diagram of corrosion process of Ni under the electrolyte with both Cl⁻ and Br⁻ (Fig. R25).

Fig. R24 a Potentiostatic polarization curves of Ni foil tested in ClE, BrE, and ClBrE, respectively. **b** Corresponding morphology of pits on Ni foil corroded by Cl⁻, Br⁻, and both Cl⁻ and Br⁻ during the potentiostatic polarization tests.

Fig. R25 a, b, c Schematic illustrations of Ni foil corroded by Cl^- , Br^- , and both Cl^- and Br^- , respectively. Insets are detailed views of NiO passive film corroded by halide.

As the replenishment of seawater, in addition to the increase of Br^- content, the concentrations of Cl^- would also increase, which was harmful to the anodes and might affect their corrosion behaviors. Therefore, the corrosion behaviors of anodes in high salt concentrations were also investigated. We found that **Br^- had a non-negligible influence on the corrosion behavior of anode even if the Cl^- concentration was increased by 5-fold (2.5 M)**. As shown in Fig. R26b, the lifetime of NiFe-H anode was drastically reduced in 1 M NaOH + 2 M NaCl + 0.5 M NaBr electrolyte, indicating the harmful corrosion effect of Br^- . The morphologies of NiFe-H anodes after the 2 h durability test in these two electrolytes were observed. In 1 M NaOH + 2.5 M NaCl electrolyte, only some pits appeared on the surface of anode (Fig. R26c), while in 1 M NaOH + 2 M NaCl + 0.5 M NaBr electrolyte, the spalling of catalyst layer accompanied by some pits on the Ni substrate can be observed (Fig. R26d).

Thus, even at high salt concentrations, the corrosion behavior of anode in chloride and bromide was similar to that of low salt concentrations.

Fig. R16 a LSV curves of NiFe-H measured in 1 M NaOH + 2.5 M NaCl and 1 M NaOH + 2 M NaCl + 0.5 M NaBr electrolytes, respectively. The scan rates were 5 mV s⁻¹. b The corresponding durability of NiFe-H at a current density of 400 mA cm⁻². c, d Corresponding morphology of NiFe-H observed after 2 h operations.

1.2 Catalyst layers

The corrosion behaviors of electrode with **different morphologies** were studied. We synthesized the NiFe-LDH nanoarrays electrode with different morphologies by electrodeposition (NiFe-E, Fig. R27a) and hydrothermal method (NiFe-H, Fig. R27d). Their corrosion behaviors were studied in 1 M NaOH + 0.5 M NaCl and 1 M NaOH + 0.5 M NaBr electrolytes. The results demonstrated that **NiFe-E and NiFe-H exhibited similar corrosion behaviors in the same electrolyte**, where Cl⁻ led to the formation of pits while Br⁻ caused the exfoliation of catalyst layer.

Fig. R27 a, d SEM morphologies of NiFe-E and NiFe-H. **b, c** SEM morphologies of NiFe-E observed after 20 h operation in 1 M NaOH + 0.5 M NaCl and in 1 M NaOH + 0.5 M NaBr. **e, f** SEM morphologies of NiFe-H observed after 48 h operation in 1 M NaOH + 0.5 M NaCl and in 1 M NaOH + 0.5 M NaBr.

Furthermore, the corrosion behaviors of anodes with different catalyst layers were investigated. Here, we chose the typical OER electrocatalysts including NiCo-LDH nanoarrays/Ni foam (NiCo-LDH) and phosphorus-doped NiFe-LDH nanoarrays/Ni foam (NiFeP) anodes as the study specimen. The results (Fig. R28 and R29) revealed that the corrosion behaviors of NiCo-LDH and NiFeP in Br^- and Cl^- were comparable to that of the NiFe-E electrode, indicating that **Br^- corrosion induced spalling of the catalyst layer was a common phenomenon in integrated electrodes (substrate + catalyst on surface).**

Fig. R28 a The LSV curves of NiCo-LDH anode measured in 1 M NaOH + 0.5 M NaCl and 1 M NaOH + 0.5 M NaBr electrolytes. The scan rate was 5 mV s^{-1} . **b** The corresponding durability tests of NiCo-LDH at a current density of 200 mA cm^{-2} . **c, d, e** Corresponding SEM morphologies of original NiCo-LDH and morphologies observed after 24 h operation in 1 M NaOH + 0.5 M NaCl and in 1 M NaOH + 0.5 M NaBr.

Fig. R29 a The LSV curves of NiFeP anode measured in 1 M NaOH + 0.5 M NaCl and 1 M NaOH + 0.5 M NaBr electrolytes. The scan rate was 5 mV s⁻¹. **b** The corresponding durability tests of NiFeP at a current density of 400 mA cm⁻². **c, d, e** Corresponding SEM morphologies of original NiFeP and morphologies observed after 70 h operation in 1 M NaOH + 0.5 M NaCl and in 1 M NaOH + 0.5 M NaBr.

Based on the above results, we can draw two conclusions: 1) Br⁻ corrosion induced spalling of the catalyst layer was a **common phenomenon in integrated electrodes** (substrate + catalyst on surface). 2) **Even though Br⁻ is of trace amount** (0.53 mM) in seawater, its influence on the corrosion behavior of Ni substrates should be attracted more attention. Thus, **it is necessary to investigate the effect of Br⁻ on the corrosion of anodes.**

2. Mechanism analysis

As for mechanism analysis, we supplemented the three pitting initiation mechanisms, including (a) adsorption-induced mechanism, (b) ion migration and penetration model, and (c) mechanical film breakdown theory (Fig. R30).¹⁻⁹ A detailed description of these mechanisms is as follows.

(a) Adsorption-induced mechanism. When the voltage is applied to anodes, halide ions will adsorb on the anode surface and then replace the surface oxygen of the passive layer, leading to the formation of a complex with metal ions. Then the complex will dissolve and diffuse into the solution to form the hydroxide. Specifically, the passivation rate is faster than corrosion rate ($i_{\text{passivation}} > i_{\text{pitting}}$) at the beginning, the passive layer becomes thicker. When the corrosion rate increases and becomes larger than passivation rate ($i_{\text{pitting}} > i_{\text{passivation}}$), the passive layer will become thinner at the corrosion sites. As the corrosion rate continues to increase and becomes much higher than passivation rate ($i_{\text{pitting}} \gg i_{\text{passivation}}$), the breakage of passive film and the corrosion of the underneath substrate will happen.

(b) Ion migration and penetration model. Due to the influence of the high electric field, halide ions will penetrate and contaminate the oxide film, causing higher electrical and ionic conductivity along the penetration paths. During the penetration process, some vacancies may also form in the passive film. Thereby the rapid release of cation at the film-solution interface or the accumulation of vacancies at the metal-film interface may happen. Finally, the passive film will break down and followed by the corrosion of substrate.

(c) Mechanical film breakdown theory. Some mechanical factors, including electrostriction stress, blistering, and micro-capillary formation, may lead to the strain on the passive film. At

some weak sites, the stress on the film may exceed the mechanical breakdown stress, causing the formation of cracks in the film. Then the film will rupture rapidly and form a channel where the halide ions can pass through to corrode the underneath metal.

Note that the mechanical film breakdown usually caused by mechanical factors, which was less relevant to the halide species. Thus, we consider that the pitting initiation of Ni substrates in CIE and BrE mainly involves **a combination of adsorption and permeation mechanisms**. Therefore, **we conducted the density functional theory simulations to study the adsorption and carried out the nudged elastic band calculations to investigate the diffusion**.

Fig. R30 a, b, c Schematic illustrations of three mechanisms for pit initiation, including adsorption-induced mechanism, ion migration and penetration model, and mechanical film breakdown theory.

3. Data processing and analysis

As for EIS analysis, to obtain a more accurate value of CPE-n (Fig. 2c in the previous manuscript) from time-dependent Bode plots (Bode-phase and Bode-amplitude plots), the graphical method^{10,11} was adopted. We plotted the imaginary part of the impedance-frequency plots (Fig. R31), where the slope at high frequency was $-\alpha$ (i.e., CPE-n). Therefore, by fitting the slope of the high-frequency region, we obtained the variation of α with cycles, and the results were shown in Fig. R32. As the number of cycles increased, **α of Ni foil in CIE decreased gradually to less than 0.5, suggesting the formation of pitting pores**. In contrast, **α of Ni foil in BrE were maintained, indicating that the electrode surface became rough due to corrosion**.

Fig. R31 a, b, d, e Time-dependent Bode-phase and Bode-amplitude plots of Ni foil in CIE and BrE, respectively. c, f Corresponding imaginary part of the impedance-frequency plots.

Fig. R32 α -cycle plots obtained by fitting the slope of the high frequency region in time-dependent imaginary part of the impedance-frequency plots.

4. Mistakes correction and English revision

Some mistakes, such as the wrong sketch of EC, inaccurate definition explanation of R_p , and wrong unit of Z' , etc. were corrected, which were highlighted in the manuscript and Supplementary Information. In addition, experimental details of the SVET test were added to complete. Finally, we have modified the English language.

In a word, after revision and experimental additions as above, we believed that our point was sufficiently accepted and also **provided significant theoretical guidance and research directions for the future design of electrodes for seawater electrolysis**. Thus, we thought our paper was innovative enough to be published in Nature Communications.

References:

- 1 Sharma, S. K. *Green corrosion chemistry and engineering* (Wiley - VCH Verlag GmbH & Co. KGaA, 2011).
 - 2 Frankel, G. S. Pitting corrosion of metals: a review of the critical factors. *J. Electrochem. Soc.* **145**, 2186-2198 (1998).
 - 3 Soltis, J. Passivity breakdown, pit initiation and propagation of pits in metallic materials - review. *Corros. Sci.* **90**, 5-22 (2015).
 - 4 Kolotyrkin, J. M. Effects of anions on the dissolution kinetics of metals. *J. Electrochem. Soc.* **108**, 209 (1961).
 - 5 Zhang, B. et al. Unmasking chloride attack on the passive film of metals. *Nat. Commun.* **9**, 2559 (2018).
 - 6 Marcus, P., Maurice, V. & Strehblow, H. H. Localized corrosion (pitting): a model of passivity breakdown including the role of the oxide layer nanostructure. *Corros. Sci.* **50**, 2698-2704 (2008).
 - 7 Hoar, T. P. The production and breakdown of the passivity of metals. *Corros. Sci.* **7**, 341-355 (1967).
 - 8 Schmuki, P. From bacon to barriers: a review on the passivity of metals and alloys. *J. Solid State Electrochem.* **6**, 145-164 (2014).
 - 9 Burstein, G. T., Liu, C., Souto, R. M. & Vines, S. P. Origins of pitting corrosion. *Corros. Eng. Sci. Techn.* **39**, 25-30 (2013).
 - 10 Orazem, M. E., Pébère, N. & Tribollet, B. Enhanced graphical representation of electrochemical impedance data. *J. Electrochem. Soc.* **153**, B129 (2006).
 - 11 Jorcin, J., Orazem, M. E., Pébère, N. & Tribollet, B. CPE analysis by local electrochemical impedance spectroscopy. *Electrochim. Acta* **51**, 1473-1479 (2006).
-

REVIEWERS' COMMENTS

Reviewer #1 (Remarks to the Author):

All the issues I raised have been properly addressed by the authors. I believe it is suitable for publication now.

Reviewer #2 (Remarks to the Author):

The Authors have provided a detailed and quite comprehensive reply to the comments and criticisms made by the Reviewers, particularly in respect to those from my side. I have also observed how the manuscript is greatly improved by including the necessary corrections as well as improving its readability.

Unfortunately, I maintain one issue of disagreement in regards to the graphical presentation of the EIS data throughout the work, since the authors maintain their initial decision of not plotting the simulated data -after using their EC analysis- together with the experimental data.

Their rationale for not plotting the simulated data is that no complete simulation of the impedance data could be achieved although trying many EC configurations, which is mainly of relevance in the low frequency limit of the spectra. But I consider that failure not to be a drawback, but readers should be aware of the limits of application of the analysis and the lack of satisfactory modelling at that low frequency limit, and this can only be made by plotting together the simulated and experimental data. The occurrence of such discrepancies is understandable for the complex electrochemical system involved, especially when pitting may be occurring even at a metastable stage, and the work can be accepted with that limitation.

On the other hand, not showing the simulated data may lead to a wrong conclusion that the proposed EC simulates the complete frequency spectra described in the experimental section without limitations, which is certainly not the case.

Reviewer #3 (Remarks to the Author):

In the feedback comments, the authors provided three detailed mechanisms for pitting corrosion initiation and a schematic diagram of corrosion process of Ni under the electrolyte with both Cl⁻ and Br⁻, and explained the corrosion processes of Ni in the electrolyte with both Cl⁻ and Br⁻. These fully demonstrate that the influence of Br⁻ on anodic corrosion behavior cannot be ignore. In summary, I suggest accepting this revised work.

Response to Reviewer 2

The Authors have provided a detailed and quite comprehensive reply to the comments and criticisms made by the Reviewers, particularly in respect to those from my side. I have also observed how the manuscript is greatly improved by including the necessary corrections as well as improving its readability.

Unfortunately, I maintain one issue of disagreement in regards to the graphical presentation of the EIS data throughout the work, since the authors maintain their initial decision of not plotting the simulated data -after using their EC analysis- together with the experimental data.

Their rationale for not plotting the simulated data is that no complete simulation of the impedance data could be achieved although trying many EC configurations, which is mainly of relevance in the low frequency limit of the spectra. But I consider that failure not to be a drawback, but readers should be aware of the limits of application of the analysis and the lack of satisfactory modelling at that low frequency limit, and this can only be made by plotting together the simulated and experimental data. The occurrence of such discrepancies is understandable for the complex electrochemical system involved, especially when pitting may be occurring even at a metastable stage, and the work can be accepted with that limitation.

On the other hand, not showing the simulated data may lead to a wrong conclusion that the proposed EC simulates the complete frequency spectra described in the experimental section without limitations, which is certainly not the case.

Response: We thank the reviewer for your precious comments and have revised our manuscript according to your suggestions.

We have added the fitted data to the potential-dependent (Fig.R1) and time-dependent Bode plots (Fig.R2). The corresponding results and discussion have been added in the Revised Manuscript and Supplementary Information (Fig. 1 and Supplementary Fig.15). In addition, the lack of satisfactory EC model modelling at the low frequency of impedance spectra and the limitations of the application of this method are also clarified in the Revised Supplementary Information.

“We used the EC shown in Supplementary Fig.13a to quantify the α and investigate the change in the surface morphology of anode during the corrosion process (Supplementary Table 4). However, the fitting curves also did not match very well in the low-frequency region. Thus, in order to obtain an accurate value of α , the graphical method was adopted.”

Fig. R1 Anti-pitting ability assessments of Ni substrates in CIE and BrE. a CPCs of Ni substrates tested in CIE and BrE at a scan rate of 10 mV s⁻¹. **b, c** Comparison of E_b , E_{pit} , and Tafel slope_{pit} values obtained from CPCs. **d, e** Bode-phase plots of Ni foil at different potentials in CIE and BrE, respectively. **f** Corresponding equivalent resistances (R_{p1} and R_{p2}) and potentials of Ni foil at different potentials in two electrolytes, where R_{p1} and R_{p2} are the polarization resistances at different frequency regions. The individual equivalent circuit models for different potentials are embedded in the diagram.

Fig. R2 a, b, d, e Time-dependent Bode-phase and Bode-amplitude plots of Ni foil in CIE and BrE, respectively. **e, f** Corresponding imaginary part of the impedance-frequency plots.